# All Points Matter: Entropy-Regularized Distribution Alignment for Weakly-supervised 3D Segmentation

**Liyao Tang[1], Zhe Chen[2], Shanshan Zhao[1], Chaoyue Wang[1], Dacheng Tao[1]**
[1] The University of Sydney, Australia [2] La Trobe University, Australia
ltan9687@uni.sydney.edu.au, zhe.chen@latrobe.edu.au
chaoyue.wang@outlook.com, {sshan.zhao00, dacheng.tao}@gmail.com

## Abstract

Pseudo-labels are widely employed in weakly supervised 3D segmentation tasks where only sparse ground-truth labels are available for learning. Existing methods often rely on empirical label selection strategies, such as confidence thresholding, to generate beneficial pseudo-labels for model training. This approach may, however, hinder the comprehensive exploitation of unlabeled data points. We hypothesize that this selective usage arises from the noise in pseudo-labels generated on unlabeled data. The noise in pseudo-labels may result in significant discrepancies between pseudo-labels and model predictions, thus confusing and affecting the model training greatly. To address this issue, we propose a novel learning strategy to regularize the generated pseudo-labels and effectively narrow the gaps between pseudo-labels and model predictions. More specifically, our method introduces an Entropy Regularization loss and a Distribution Alignment loss for weakly supervised learning in 3D segmentation tasks, resulting in an ERDA learning strategy. Interestingly, by using KL distance to formulate the distribution alignment loss, it reduces to a deceptively simple cross-entropy-based loss which optimizes both the pseudo-label generation network and the 3D segmentation network simultaneously. Despite the simplicity, our method promisingly improves the performance. We validate the effectiveness through extensive experiments on various baselines and large-scale datasets. Results show that ERDA effectively enables the effective usage of all unlabeled data points for learning and achieves state-of-the-art performance under different settings. Remarkably, our method can outperform fully-supervised baselines using only 1% of true annotations. Code and model will be made publicly available at https://github.com/LiyaoTang/ERDA.

## 1 Introduction

Point cloud semantic segmentation is a crucial task for 3D scene understanding and has various applications [27, 83], such as autonomous driving, unmanned aerial vehicles, and augmented reality. Current state-of-the-art approaches heavily rely on large-scale and densely annotated 3D datasets, which are costly to obtain [21, 33]. To avoid demanding exhaustive annotation, weakly-supervised point cloud segmentation has emerged as a promising alternative. It aims to leverage a small set of annotated points while leaving the majority of points unlabeled in a large point cloud dataset for learning. Although current weakly supervised methods can offer a practical and cost-effective way to perform point cloud segmentation, their performance is still sub-optimal compared to fully-supervised approaches.

During the exploration of weak supervision [85], a significant challenge is the insufficient training signals provided by the highly sparse labels. To tackle this issue, pseudo-labeling methods [85, 94, 32] have been proposed, which leverage predictions on unlabeled points as labels to facilitate the learning of the segmentation network. Despite some promising results, these pseudo-label methods have been outperformed by some recent methods based on consistency regularization [47, 87]. We tend to attribute this to the use of label selection on pseudo-labels, such as confidence thresholding, which

37th Conference on Neural Information Processing Systems (NeurIPS 2023).

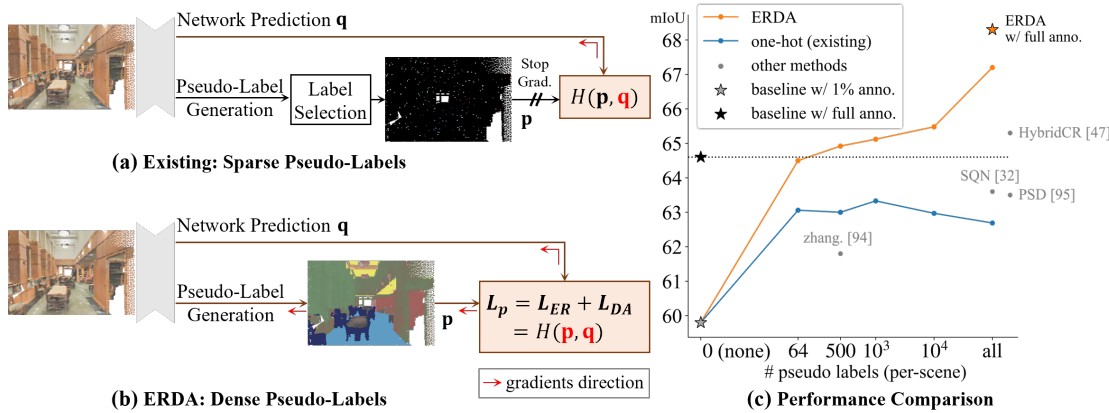

Figure 1. While existing pseudo-labels (a) are limited in the exploitation of unlabeled points, ERDA (b) simultaneously optimizes the pseudo-labels **p** and predictions **q** taking the same and simple form of cross-entropy. By reducing the noise via entropy regularization and bridging their distributional discrepancies, ERDA produces informative pseudo-labels that neglect the need for label selection. As in (c), it thus enables the model to consistently benefit from more pseudo-labels, surpasses other methods and its fully-supervised baseline, and can be extended to advance the fully-supervised performance.

could lead to unlabeled points being wasted and under-explored. We hypothesize that the need for label selection arises from the low-confidence pseudo-labels assigned to unlabeled points, which are known for their noises [21] and potential unintended bias [60, 100]. These less reliable and noisy pseudo-labels could contribute to discrepancies between the pseudo-labels and the model predictions, which might confuse and impede the learning process to a great extent.

By addressing the above problem for label selection in weakly supervised 3D segmentation, we propose a novel learning-based approach in this study. Our method aims to leverage the information from all unlabeled points by mitigating the negative effects of the noisy pseudo-labels and the distributional discrepancy.

Specifically, we introduce two learning objectives for the pseudo-label generation process. Firstly, we introduce an *entropy regularization* (ER) objective to reduce the noise and uncertainty in the pseudo-labels. This regularization promotes more informative, reliable, and confident pseudo-labels, which helps alleviate the limitations of noisy and uncertain pseudo-labels. Secondly, we propose a *distribution alignment* (DA) loss that minimizes statistical distances between pseudo-labels and model predictions. This ensures that the distribution of generated pseudo-labels remains close to the distribution of model predictions when regularizing their entropy.

In particular, we discover that formulating the distribution alignment loss using KL distance enables a simplification of our method into a cross-entropy-style learning objective that optimizes both the pseudo-label generator and the 3D segmentation network simultaneously. This makes our method straightforward to implement and apply. By integrating the entropy regularization and distribution alignment, we achieve the ERDA learning strategy, as shown in Fig. 1.

Empirically, we comprehensively experiment with three baselines and different weak supervision settings, including 0.02% (1-point, or 1pt), 1%, and 10%. Despite its concise design, our ERDA outperforms existing weakly-supervised methods on large-scale point cloud datasets such as S3DIS [2], ScanNet [16], and SensatUrban [33]. Notably, our ERDA can surpass the fully-supervised baselines using only 1% labels, demonstrating its significant effectiveness in leveraging pseudo-labels. Furthermore, we validate the scalability of our method by successfully generalizing it to more other settings, which illustrates the benefits of utilizing dense pseudo-label supervision with ERDA.

## 2   Related Work

**Point cloud segmentation.** Point cloud semantic segmentation aims to assign semantic labels to 3D points. The cutting-edge methods are deep-learning-based and can be classified into projection-based and point-based approaches. Projection-based methods project 3D points to grid-like structures, such as 2D image [84, 55, 39, 12, 4, 45] or 3D voxels [15, 71, 67, 28, 22, 23, 76]. Alternatively, point-based

methods directly operate on 3D points [56, 57]. Recent efforts have focused on novel modules and backbones to enhance point features, such as 3D convolution [3, 48, 78, 68, 51, 58], attentions [34, 26, 97, 79, 42, 59], graph-based methods [74, 44], and other modules such as sampling [18, 86, 88, 7] and post-processing [54, 35, 66]. Although these methods have made significant progress, they rely on large-scale datasets with point-wise annotation and struggle with few labels [85]. To address the demanding requirement of point-wise annotation, our work explores weakly-supervised learning for 3D point cloud segmentation.

**Weakly-supervised point cloud segmentation.** Compared to weakly-supervised 2D image segmentation [99, 49, 75, 1, 64], weakly-supervised 3D point cloud segmentation is less explored. In general, weakly-supervised 3D segmentation task focus on highly sparse labels: only a few scattered points are annotated in large point cloud scenes. Xu and Lee [85] first propose to use 10x fewer labels to achieve performance on par with a fully-supervised point cloud segmentation model. Later studies have explored more advanced ways to exploit different forms of weak supervision [77, 14, 40] and human annotations [53, 69]. Recent methods tend to introduce perturbed self-distillation [95], consistency regularization [85, 62, 80, 81, 43], and leverage self-supervised learning [62, 37, 47, 87] based on contrastive learning [29, 10]. Pseudo-labels are another approach to leverage unlabeled data, with methods such as pre-training networks on colorization tasks [94], using iterative training [32], employing separate networks to iterate between learning pseudo-labels and training 3D segmentation networks [53], or using super-point graph [44] with graph attentional module to propagate the limited labels over super-points [13]. However, these existing methods often require expensive training due to hand-crafted 3D data augmentations [95, 87, 80, 81], iterative training [53, 32], or additional modules [87, 32], complicating the adaptation of backbone models from fully-supervised to weakly-supervised learning. In contrast, our work aims to achieve effective weakly supervised learning for the 3D segmentation task with straightforward motivations and simple implementation.

**Pseudo-label refinement.** Pseudo-labeling [46], a versatile method for entropy minimization [24], has been extensively studied in various tasks, including semi-supervised 2D classification [82, 60], segmentation [64, 89], and domain adaptation [98, 73]. To generate high-quality supervision, various label selection strategies have been proposed based on learning status [72, 91, 20], label uncertainty [60, 98, 73, 50], class balancing [100], and data augmentations [64, 89, 100]. Our method is most closely related to the works addressing bias in supervision, where mutual learning [20, 70, 92] and distribution alignment [100, 31, 41] have been discussed. However, these works typically focus on class imbalance [100, 31] and rely on iterative training [70, 20, 92, 41], label selection [20, 31], and strong data augmentations [100, 31], which might not be directly applicable to 3D point clouds. For instance, common image augmentations [64] like cropping and resizing may translate to point cloud upsampling [96], which remains an open question in the related research area. Rather than introducing complicated mechanisms, we argue that proper regularization on pseudo-labels and its alignment with model prediction can provide significant benefits using a very concise learning approach designed for the weakly supervised 3D point cloud segmentation task.

Besides, it is shown that the data augmentations and repeated training in mutual learning [70, 38] are important to avoid the feature collapse, *i.e.,* the resulting pseudo-labels being uniform or the same as model predictions. We suspect the cause may originate from the entropy term in their use of raw statistical distance by empirical results, which potentially matches the pseudo-labels to noisy and confusing model prediction, as would be discussed in Sec. 3.2. Moreover, in self-supervised learning based on clustering [5] and distillation [6], it has also been shown that it would lead to feature collapse if matching to a cluster assignment or teacher output of a close-uniform distribution with high entropy, which agrees with the intuition in our ER term.

# 3 Methodology

## 3.1 Formulation of ERDA

As previously mentioned, we propose the ERDA approach to alleviate noise in the generated pseudo-labels and reduce the distribution gaps between them and the segmentation network predictions. In general, our ERDA introduces two loss functions, including the entropy regularization loss and the distribution alignment loss for the learning on pseudo-labels. We denote the two loss functions as $L_{ER}$ and $L_{DA}$, respectively. Then, we have the overall loss of ERDA as follows:

$$L_p = \lambda L_{ER} + L_{DA}, \tag{1}$$

where the $\lambda > 0$ modulates the entropy regularization which is similar to the studies [46, 24].

Before detailing the formulation of $L_{ER}$ and $L_{DA}$, we first introduce the notation. While the losses are calculated over all unlabeled points, we focus on one single unlabeled point for ease of discussion. We denote the pseudo-label assigned to this unlabeled point as $\mathbf{p}$ and the corresponding segmentation network prediction as $\mathbf{q}$. Each $\mathbf{p}$ and $\mathbf{q}$ is a 1D vector representing the probability over classes.

**Entropy Regularization loss.** We hypothesize that the quality of pseudo-labels can be hindered by noise, which in turn affects model learning. Specifically, we consider that the pseudo-label could be more susceptible to containing noise when it fails to provide a confident pseudo-labeling result, which leads to the presence of a high-entropy distribution in $\mathbf{p}$.

To mitigate this, for the $\mathbf{p}$, we propose to reduce its noise level by minimizing its Shannon entropy, which also encourages a more informative labeling result [61]. Therefore, we have:

$$L_{ER} = H(\mathbf{p}), \tag{2}$$

where $H(\mathbf{p}) = \sum_i -p_i \log p_i$ and $i$ iterates over the vector. By minimizing the entropy of the pseudo-label as defined above, we promote more confident labeling results to help resist noise in the labeling process[1].

**Distribution Alignment loss.** In addition to the noise in pseudo-labels, we propose that significant discrepancies between the pseudo-labels and the segmentation network predictions could also confuse the learning process and lead to unreliable segmentation results. In general, the discrepancies can stem from multiple sources, including the noise-induced unreliability of pseudo-labels, differences between labeled and unlabeled data [100], and variations in pseudo-labeling methods and segmentation methods [92, 20]. Although entropy regularization could mitigate the impact of noise in pseudo-labels, significant discrepancies may still persist between the pseudo-labels and the predictions of the segmentation network. To mitigate this issue, we propose that the pseudo-labels and network can be jointly optimized to narrow such discrepancies, making generated pseudo-labels not diverge too far from the segmentation predictions. Therefore, we introduce the distribution alignment loss.

To properly define the distribution alignment loss ($L_{DA}$), we measure the KL divergence between the pseudo-labels ($\mathbf{p}$) and the segmentation network predictions ($\mathbf{q}$) and aim to minimize this divergence. Specifically, we define the distribution alignment loss as follows:

$$L_{DA} = KL(\mathbf{p}||\mathbf{q}), \tag{3}$$

where $KL(\mathbf{p}||\mathbf{q})$ refers to the KL divergence. Using the above formulation has several benefits. For example, the KL divergence can simplify the overall loss $L_p$ into a deceptively simple form that demonstrates desirable properties and also performs better than other distance measurements. More details will be presented in the following sections.

**Simplified ERDA.** With the $L_{ER}$ and $L_{DA}$ formulated as above, given that $KL(\mathbf{p}||\mathbf{q}) = H(\mathbf{p}, \mathbf{q}) - H(\mathbf{p})$ where $H(\mathbf{p}, \mathbf{q})$ is the cross entropy between $\mathbf{p}$ and $\mathbf{q}$, we can have a simplified ERDA formulation as:

$$L_p = H(\mathbf{p}, \mathbf{q}) + (\lambda - 1)H(\mathbf{p}). \tag{4}$$

In particular, when $\lambda = 1$, we obtain the final ERDA loss[2]:

$$L_p = H(\mathbf{p}, \mathbf{q}) = \sum_i -p_i \log q_i \tag{5}$$

The above simplified ERDA loss describes that the entropy regularization loss and distribution alignment loss can be represented by a single cross-entropy-based loss that optimizes both $\mathbf{p}$ and $\mathbf{q}$.

We would like to emphasize that Eq. (5) is distinct from the conventional cross-entropy loss. The conventional cross-entropy loss utilizes a fixed label and only optimizes the term within the logarithm function, whereas the proposed loss in Eq. (5) optimizes both $\mathbf{p}$ and $\mathbf{q}$ simultaneously.

---

[1]We note that our entropy regularization aims for entropy *minimization* on pseudo-labels; and we consider noise as the uncertain predictions by the pseudo-labels, instead of incorrect predictions.

[2]We would justify the choice of $\lambda$ in the following Sec. 3.2 as well as Sec. 4.3

| $L_{DA}$ | $KL(\mathbf{p}\|\mathbf{q})$ | $KL(\mathbf{q}\|\mathbf{p})$ | $JS(\mathbf{p},\mathbf{q})$ | $MSE(\mathbf{p},\mathbf{q})$ |
|---|---|---|---|---|
| $L_p$ | $H(\mathbf{p},\mathbf{q}) - (1-\lambda)H(\mathbf{p})$ | $H(\mathbf{q},\mathbf{p}) - H(\mathbf{q}) + \lambda H(\mathbf{p})$ | $H(\frac{\mathbf{p}+\mathbf{q}}{2}) - (\frac{1}{2}-\lambda)H(\mathbf{p}) - \frac{1}{2}H(\mathbf{q})$ | $\frac{1}{2}\sum_i (p_i - q_i)^2 + \lambda H(\mathbf{p})$ |
| **S1** | $0$ | $q_i - \mathbb{1}_{k=i}$ | $0$ | $0$ |
| **S2** | $(\lambda-1)p_i \sum_j p_j \log \frac{p_i}{p_j}$ | $\frac{1}{K} - p_i + \lambda p_i \sum_j p_j \log \frac{p_i}{p_j}$ | $p_i \sum_{j\neq i} p_j(\frac{1}{2}\log \frac{Kp_i+1}{Kp_j+1} + (\lambda - \frac{1}{2})\log \frac{p_i}{p_j})$ | $-p_i^2 + p_i \sum_j p_j^2 + \lambda p_i \sum_j p_j \log \frac{p_i}{p_j}$ |

Table 1. The formulation of $L_p$ using different functions to formulate $L_{DA}$. We study the gradient update on $s_i$, *i.e.,* $-\frac{\partial L_p}{\partial s_i}$ under different situations. **S1**: update given confident pseudo-label, $\mathbf{p}$ being one-hot with $\exists p_k \in \mathbf{p}, p_k \to 1$. **S2**: update given confusing prediction, $\mathbf{q}$ being uniform with $q_1 = ... = q_K = \frac{1}{K}$. More analysis as well as visualization can be found in the Sec. 3.2 and the supplementary Appendix C.

## 3.2 Delving into the Benefits of ERDA

To formulate the distribution alignment loss, different functions can be employed to measure the differences between $\mathbf{p}$ and $\mathbf{q}$. In addition to the KL divergence, there are other distance measurements like mean squared error (MSE) or Jensen-Shannon (JS) divergence for replacement. Although many mutual learning methods [20, 92, 41, 38] have proven the effectiveness of KL divergence, a detailed comparison of KL divergence against other measurements is currently lacking in the literature. In this section, under the proposed ERDA learning framework, we show by comparison that $KL(\mathbf{p}||\mathbf{q})$ is a better choice and ER is necessary for weakly-supervised 3D segmentation.

To examine the characteristics of different distance measurements, including $KL(\mathbf{p}||\mathbf{q})$, $KL(\mathbf{q}||\mathbf{p})$, $JS(\mathbf{p}||\mathbf{q})$, and $MSE(\mathbf{p}||\mathbf{q})$, we investigate the form of our ERDA loss $L_p$ and its impact on the learning for pseudo-label generation network given two situations during training.

More formally, we shall assume a total of $K$ classes and define that a pseudo-label $\mathbf{p} = [p_1, ..., p_K]$ is based on the confidence scores $\mathbf{s} = [s_1, ..., s_K]$, and that $\mathbf{p} = \text{softmax}(\mathbf{s})$. Similarly, we have a segmentation network prediction $\mathbf{q} = [q_1, ..., q_K]$ for the same point. We re-write the ERDA loss $L_p$ in various forms and investigate the learning from the perspective of gradient update, as in Tab. 1.

**Situation 1: Gradient update given confident pseudo-label $\mathbf{p}$.** We first specifically study the case when $\mathbf{p}$ is very certain and confident, *i.e.,* $\mathbf{p}$ approaching a one-hot vector. As in Tab. 1, most distances yield the desired zero gradients, which thus retain the information of a confident and reliable $\mathbf{p}$. In this situation, however, the $KL(\mathbf{q}||\mathbf{p})$, rather than $KL(\mathbf{p}||\mathbf{q})$ in our method, produces non-zero gradients that would actually increase the noise among pseudo-labels during its learning, which is not favorable according to our motivation.

**Situation 2: Gradient update given confusing prediction $\mathbf{q}$.** In addition, we are also interested in how different choices of distance and $\lambda$ would impact the learning on pseudo-label if the segmentation model produces confusing outputs, *i.e.,* $\mathbf{q}$ tends to be uniform. In line with the motivation of ERDA learning, we aim to regularize the pseudo-labels to mitigate potential noise and bias, while discouraging uncertain labels with little information. However, as in Tab. 1, most implementations yield non-zero gradient updates to the pseudo-label generation network. This update would make $\mathbf{p}$ closer to the confused $\mathbf{q}$, thus increasing the noise and degrading the training performance. Conversely, only $KL(\mathbf{p}||\mathbf{q})$ can produce a zero gradient when integrated with the entropy regularization with $\lambda = 1$. That is, only ERDA in Eq. (5) would not update the pseudo-label generation network when $\mathbf{q}$ is not reliable, which avoids confusing the $\mathbf{p}$. Furthermore, when $\mathbf{q}$ is less noisy but still close to a uniform vector, it is indicated that there is a large close-zero plateau on the gradient surface of ERDA, which benefits the learning on $\mathbf{p}$ by resisting the influence of noise in $\mathbf{q}$.

In addition to the above cases, the gradients of ERDA in Eq. (5) could be generally regarded as being aware of the noise level and the confidence of both pseudo-label $\mathbf{p}$ and the corresponding prediction $\mathbf{q}$. Especially, ERDA produces larger gradient updates on noisy pseudo-labels, while smaller updates on confident and reliable pseudo-labels or given noisy segmentation prediction. Therefore, our formulation demonstrates its superiority in fulfilling our motivation of simultaneous noise reduction and distribution alignment, where both $L_{ER}$ and KL-based $L_{DA}$ are necessary. We provide more empirical studies in ablation (Sec. 4.3) and detailed analysis in the supplementary.

## 3.3 Implementation Details on Pseudo-Labels

In our study, we use a prototypical pseudo-label generation process due to its popularity as well as simplicity [94]. Specifically, prototypes [63] denote the class centroids in the feature space, which

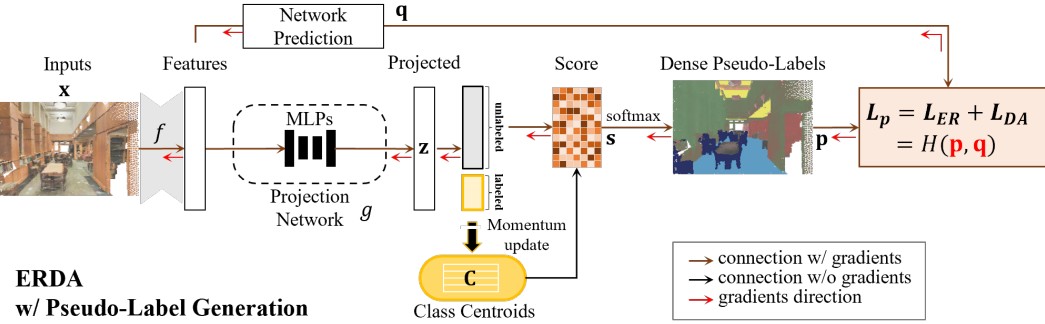

Figure 2. Detailed illustration of our ERDA with the prototypical pseudo-label generation process, which is shared for both (a) and (b) in Fig. 1.

are calculated based on labeled data, and pseudo-labels are estimated based on the feature distances between unlabeled points and class centroids.

As shown in Fig. 2, we use a momentum-based prototypical pseudo-label generation process due to its popularity as well as simplicity [94, 85, 93]. Specifically, prototypes [63] denote the class centroids in the feature space, which are calculated based on labeled data, and pseudo-labels are estimated based on the feature distances between unlabeled points and class centroids. To avoid expensive computational costs and compromised representations for each semantic class [94, 87, 47], momentum update is utilized as an approximation for global class centroids.

Based on the momentum-updated prototypes, we attach an MLP-based projection network to help generate pseudo-labels and learn with our method. Aligned with our motivation, we do not introduce thresholding-based label selection or one-hot conversion [46, 94] to process generated pseudo-labels. More details are in the supplementary.

More formally, we take as input a point cloud $\mathcal{X}$, where the labeled points are $\mathcal{X}^l$ and the unlabeled points are $\mathcal{X}^u$. For a labeled point $\mathbf{x} \in \mathcal{X}^l$, we denote its label by $y$. The pseudo-label generation process can be described as follows:

$$\hat{C}_k = \frac{1}{N_k^l} \sum_{\mathbf{x} \in \mathcal{X}^l \wedge y=k} g \circ f(\mathbf{x}) \, , \, C_k \leftarrow mC_k + (1-m)\hat{C}_k,$$

$$\forall \mathbf{x} \in \mathcal{X}^u \, , \, s_k = d(g \circ f(\mathbf{x}), C_k) \, , \, \mathbf{p} = \mathrm{softmax}(\mathbf{s}),$$

where $C_k$ denotes the global class centroid for the $k$-th class, $N_k^l$ is the number of labeled points of the $k$-th class, $g \circ f = g(f(\cdot))$ is the transformation through the backbone network $f$ and the projection network $g$, $m$ is the momentum coefficient, and we use cosine similarity for $d(\cdot, \cdot)$ to generate the score $\mathbf{s}$. By default, we use 2-layer MLPs for the projection network $g$ and set $m = 0.999$.

Besides, due to the simplicity of ERDA, we are able to follow the setup of the baselines for training, which enables straightforward implementation and easy adaptation on various backbone models with little overhead.

**Overall objective.** Finally, with ERDA learning in Eq. (5), we maximize the same loss for both labeled and unlabeled points, segmentation task, and pseudo-label generation, where we allow the gradient to back-propagate through the (pseudo-)labels. The final loss is given as

$$L = \frac{1}{N^l} \sum_{\mathbf{x} \in \mathcal{X}^l} L_{ce}(\mathbf{q}, y) + \alpha \frac{1}{N^u} \sum_{\mathbf{x} \in \mathcal{X}^u} L_p(\mathbf{q}, \mathbf{p}), \tag{6}$$

where $L_p(\mathbf{q}, \mathbf{p}) = L_{ce}(\mathbf{q}, \mathbf{p}) = H(\mathbf{q}, \mathbf{p})$ is the typical cross-entropy loss used for point cloud segmentation, $N^l$ and $N^u$ are the numbers of labeled and unlabeled points, and $\alpha$ is the loss weight.

## 4    Experiments

We present the benefits of our proposed ERDA by experimenting with multiple large-scale datasets, including S3DIS [2], ScanNet [16], SensatUrban [33] and Pascal [19]. We also provide ablation studies for better investigation.

| settings | methods | mIoU | ceiling | floor | wall | beam | column | window | door | table | chair | sofa | bookcase | board | clutter |
|---|---|---|---|---|---|---|---|---|---|---|---|---|---|---|---|
| Fully | PointNet [56] | 41.1 | 88.8 | 97.3 | 69.8 | 0.1 | 3.9 | 46.3 | 10.8 | 59.0 | 52.6 | 5.9 | 40.3 | 26.4 | 33.2 |
| | MinkowskiNet [15] | 65.4 | 91.8 | **98.7** | 86.2 | 0.0 | 34.1 | 48.9 | 62.4 | 81.6 | **89.8** | 47.2 | 74.9 | 74.4 | 58.6 |
| | KPConv [68] | 65.4 | 92.6 | 97.3 | 81.4 | 0.0 | 16.5 | 54.5 | 69.5 | 90.1 | 80.2 | 74.6 | 66.4 | 63.7 | 58.1 |
| | SQN [32] | 63.7 | 92.8 | 96.9 | 81.8 | 0.0 | 25.9 | 50.5 | 65.9 | 79.5 | 85.3 | 55.7 | 72.5 | 65.8 | 55.9 |
| | HybridCR [47] | 65.8 | 93.6 | 98.1 | 82.3 | 0.0 | 24.4 | 59.5 | 66.9 | 79.6 | 87.9 | 67.1 | 73.0 | 66.8 | 55.7 |
| | RandLA-Net [34] | 64.6 | 92.4 | 96.8 | 80.8 | 0.0 | 18.6 | 57.2 | 54.1 | 87.9 | 79.8 | 74.5 | 70.2 | 66.2 | 59.3 |
| | **+ ERDA** | 68.3 | 93.9 | 98.5 | 83.4 | 0.0 | 28.9 | 62.6 | 70.0 | 89.4 | 82.7 | 75.5 | 69.5 | 75.3 | 58.7 |
| | CloserLook3D [51] | 66.2 | 94.2 | 98.1 | 82.7 | 0.0 | 22.2 | 57.6 | 70.4 | 91.2 | 81.2 | 75.3 | 61.7 | 65.8 | 60.4 |
| | **+ ERDA** | 69.6 | 94.5 | 98.5 | 85.2 | 0.0 | 31.1 | 57.3 | 72.2 | 91.7 | 83.6 | 77.6 | 74.8 | **75.8** | 62.1 |
| | PT [97] | 70.4 | 94.0 | 98.5 | 86.3 | 0.0 | 38.0 | **63.4** | 74.3 | 89.1 | 82.4 | 74.3 | 80.2 | 76.0 | 59.3 |
| | **+ ERDA** | **72.6** | **95.8** | 98.6 | **86.4** | 0.0 | **43.9** | 61.2 | **81.3** | **93.0** | 84.5 | **77.7** | **81.5** | 74.5 | **64.9** |
| 0.02% (1pt) | zhang *et al.* [94] | 45.8 | - | - | - | - | - | - | - | - | - | - | - | - | - |
| | PSD [95] | 48.2 | **87.9** | 96.0 | 62.1 | 0.0 | 20.6 | 49.3 | 40.9 | 55.1 | 61.9 | 43.9 | 50.7 | 27.3 | 31.1 |
| | MIL-Trans [87] | 51.4 | 86.6 | 93.2 | **75.0** | 0.0 | **29.3** | 45.3 | **46.7** | 60.5 | 62.3 | 56.5 | 47.5 | 33.7 | 32.2 |
| | HybridCR [47] | 51.5 | 85.4 | 91.9 | 65.9 | 0.0 | 18.0 | **51.4** | 34.2 | 63.8 | **78.3** | 52.4 | **59.6** | 29.9 | **39.0** |
| | RandLA-Net [34] | 40.6 | 84.0 | 94.2 | 59.0 | 0.0 | 5.4 | 40.4 | 16.9 | 52.8 | 51.4 | 52.2 | 16.9 | 27.8 | 27.0 |
| | **+ ERDA** | 48.4 | 87.3 | 96.3 | 61.9 | 0.0 | 11.3 | 45.9 | 31.7 | 73.1 | 65.1 | **57.8** | 26.1 | **36.0** | 36.4 |
| | CloserLook3D [51] | 34.6 | 33.6 | 40.5 | 52.4 | 0.0 | 21.1 | 25.4 | 35.5 | 48.9 | 48.9 | 53.9 | 23.8 | 35.3 | 30.1 |
| | **+ ERDA** | **52.0** | 90.0 | **96.7** | 70.2 | 0.0 | 21.5 | 45.8 | 41.9 | **76.0** | 65.5 | 56.1 | 51.5 | 30.6 | 30.9 |
| | PT [97] | 2.2 | 0.0 | 0.0 | 29.2 | 0.0 | 0.0 | 0.0 | 0.0 | 0.0 | 0.0 | 0.0 | 0.0 | 0.0 | 0.0 |
| | **+ ERDA** | 26.2 | 86.8 | **96.9** | 63.2 | 0.0 | 0.0 | 0.0 | 15.1 | 29.6 | 26.3 | 0.0 | 0.0 | 0.0 | 22.8 |
| 1% | zhang *et al.* [94] | 61.8 | 91.5 | 96.9 | 80.6 | 0.0 | 18.2 | 58.1 | 47.2 | 75.8 | 85.7 | 65.2 | 68.9 | 65.0 | 50.2 |
| | PSD [95] | 63.5 | 92.3 | 97.7 | 80.7 | 0.0 | 27.8 | 56.2 | 62.5 | 78.7 | 84.1 | 63.1 | 70.4 | 58.9 | 53.2 |
| | SQN [32] | 63.6 | 92.0 | 96.4 | 81.3 | 0.0 | 21.4 | 53.7 | 73.2 | 77.8 | **86.0** | 56.7 | 69.9 | 66.6 | 52.5 |
| | HybridCR [47] | 65.3 | 92.5 | 93.9 | 82.6 | 0.0 | 24.2 | **64.4** | 63.2 | 78.3 | 81.7 | 69.0 | 74.4 | 68.2 | 56.5 |
| | RandLA-Net [34] | 59.8 | 92.3 | 97.5 | 77.0 | 0.1 | 15.9 | 48.7 | 38.0 | 83.2 | 78.0 | 68.4 | 62.4 | 64.9 | 50.6 |
| | **+ ERDA** | 67.2 | 94.2 | 97.5 | 82.3 | 0.0 | 27.3 | 60.7 | 68.8 | 88.0 | 80.6 | 76.0 | 70.5 | 68.7 | 58.4 |
| | CloserLook3D [51] | 59.9 | 95.3 | **98.4** | 78.7 | 0.0 | 14.5 | 44.4 | 38.1 | 84.9 | 79.0 | 69.5 | 67.8 | 53.9 | 54.1 |
| | **+ ERDA** | 68.2 | 94.0 | 98.2 | 83.8 | 0.0 | 30.2 | 56.7 | 62.7 | **91.0** | 80.8 | 75.4 | 80.2 | 74.5 | 58.3 |
| | PT [97] | 65.8 | 94.2 | 98.2 | 83.0 | 0.0 | **44.2** | 50.4 | 68.8 | 88.1 | 83.0 | 75.2 | 47.4 | 64.3 | 59.0 |
| | **+ ERDA** | **70.4** | **95.5** | 98.1 | **85.5** | 0.0 | 30.5 | 61.7 | **73.3** | 90.1 | 82.6 | **77.6** | **80.6** | **76.0** | **63.1** |
| 10% | Xu and Lee [85] | 48.0 | 90.9 | 97.3 | 74.8 | 0.0 | 8.4 | 49.3 | 27.3 | 69.0 | 71.7 | 16.5 | 53.2 | 23.3 | 42.8 |
| | Semi-sup [37] | 57.7 | - | - | - | - | - | - | - | - | - | - | - | - | - |
| | zhang *et al.* [94] | 64.0 | - | - | - | - | - | - | - | - | - | - | - | - | - |
| | SQN [32] | 64.7 | 93.0 | 97.5 | 81.5 | 0.0 | 28.0 | 55.8 | 68.7 | 80.1 | **87.7** | 55.2 | 72.3 | 63.9 | 57.0 |
| | RandLA-Net [34] | 61.7 | 91.7 | 97.8 | 79.4 | 0.0 | 28.4 | 50.8 | 45.5 | 85.2 | 81.3 | 70.3 | 57.1 | 63.8 | 51.8 |
| | **+ ERDA** | 67.9 | 94.3 | 98.4 | 83.2 | 0.0 | 30.5 | 60.7 | 67.4 | 88.8 | 83.2 | 74.5 | 68.8 | 72.4 | 60.4 |
| | CloserLook3D [51] | 55.5 | 93.0 | 98.2 | 73.6 | 0.0 | 12.6 | 25.6 | 33.3 | 87.5 | 72.9 | 65.1 | 73.1 | 36.0 | 51.6 |
| | **+ ERDA** | 69.1 | **94.7** | **98.5** | 83.2 | 0.0 | 28.8 | 53.8 | 70.9 | **91.5** | 82.5 | 75.8 | **82.1** | **75.3** | 61.6 |
| | PT [97] | 66.0 | 93.7 | 98.3 | 83.7 | 0.0 | 35.0 | 48.1 | 70.9 | 88.3 | 81.9 | 73.2 | 60.3 | 67.3 | 57.2 |
| | **+ ERDA** | **71.7** | 94.6 | **98.5** | **86.5** | 0.0 | **49.7** | **61.3** | **82.4** | 89.8 | 84.3 | **78.0** | 70.5 | 74.1 | **62.4** |

Table 2. The results are obtained on the S3DIS datasets Area 5. For all baseline methods, we follow their official instructions in evaluation. The **bold** denotes the best performance in each setting.

## 4.1 Experimental Setup

We choose RandLA-Net [34] and CloserLook3D [51] as our primary baselines following previous works. Additionally, while transformer models [17, 52] have revolutionized the field of computer vision as well as 3D point cloud segmentation [97, 42], none of the existing works have addressed the training of transformer for point cloud segmentation with weak supervision, even though these models are known to be data-hungry [17]. We thus further incorporate the PointTransformer (PT) [97] as our baseline to study the amount of supervision demanded for effective training of transformer.

For training, we follow the setup of the baselines and set the loss weight $\alpha = 0.1$. For a fair comparison, we follow previous works [94, 95, 32] and experiment with different settings, including the 0.02% (1pt), 1% and 10% settings, where the available labels are randomly sampled according to the ratio[3]. More details are given in the supplementary.

## 4.2 Performance Comparison

**Results on S3DIS.** S3DIS [2] is a large-scale point cloud segmentation dataset that covers 6 large indoor areas with 272 rooms and 13 semantic categories. As shown in Tab. 2, ERDA significantly improves over different baselines on all settings and almost all classes. In particular, for confusing classes such as column, window, door, and board, our method provides noticeable and consistent improvements in all weak supervision settings. We also note that PT suffers from severe over-fitting and feature collapsing under the supervision of extremely sparse labels of "1pt" setting; whereas it is alleviated with ERDA, though not achieving a satisfactory performance. Such observation agrees with the understanding that transformer is data-hungry [17].

---

[3]Some super-voxel-based approaches, such as OTOC [53], additionally leverage the super-voxel partition from the dataset annotations [32]. We thus treat them as a different setting and avoid direct comparison.

| Input | Ground Truth | Baseline | ERDA | Improvement |
|-------|--------------|----------|------|-------------|

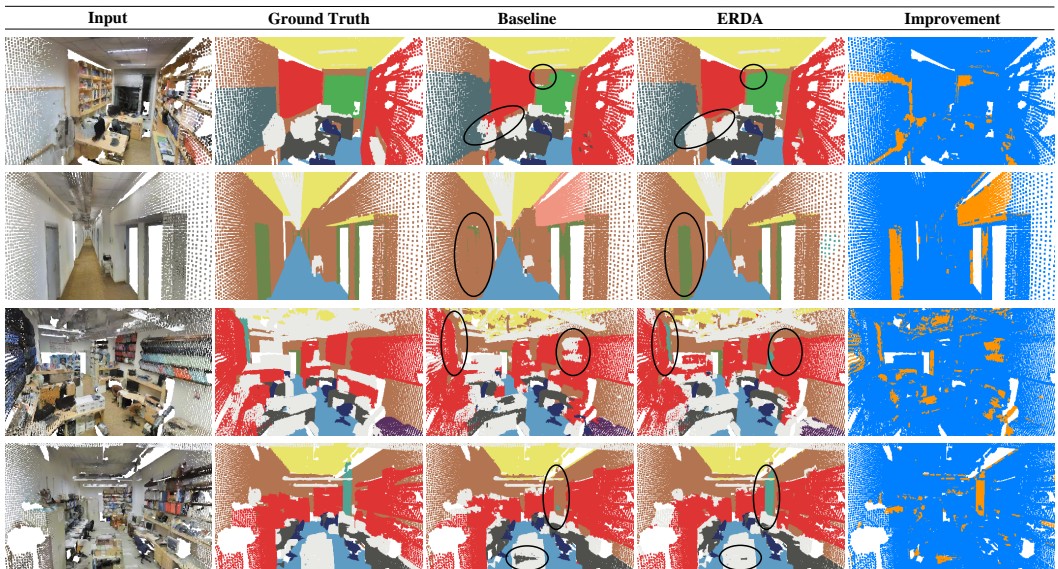

Figure 3. We show obvious improvement of our ERDA over baseline (RandLA-Net) on different scenes from S3DIS Area 5. In the office and hallway (top 2), ERDA produces more detailed and complete segmentation for windows and doors, and avoids over-expansion of the board and bookcase on the wall, thanks to the informative pseudo-labels. In more cluttered scenes (bottom 2), ERDA tends to make cleaner predictions by avoiding improper situations such as desk inside clutter and preserving important semantic classes such as columns.

| settings | methods | mIoU |
|----------|---------|------|
| Fully | PointNet [56] | 47.6 |
| | RandLA-Net [34] | 70.0 |
| | KPConv [68] | 70.6 |
| | HybridCR [47] | 70.7 |
| | PT [97] | 73.5 |
| | PointNeXt - XL [58] | 74.9 |
| | RandLA-Net **+ ERDA** | 71.0 |
| | CloserLook3D **+ ERDA** | 73.7 |
| | PT **+ ERDA** | 76.3 |
| 1% | zhang *et al.* [94] | 65.9 |
| | PSD [95] | 68.0 |
| | HybridCR [47] | 69.2 |
| | RandLA-Net **+ ERDA** | 69.4 |
| | CloserLook3D **+ ERDA** | 72.3 |
| | PT **+ ERDA** | 73.5 |

Table 3. Results on S3DIS 6-fold.

| settings | methods | mIoU |
|----------|---------|------|
| Fully | PointCNN [48] | 45.8 |
| | RandLA-Net [34] | 64.5 |
| | KPConv [68] | 68.4 |
| | HybridCR [47] | 59.9 |
| | CloserLook3D **+ ERDA** | 70.4 |
| 20pts | MIL-Trans [87] | 54.4 |
| | CloserLook3D **+ ERDA** | 57.0 |
| 0.1% | SQN [32] | 56.9 |
| | RandLA-Net **+ ERDA** | 62.0 |
| 1% | zhang *et al.* [94] | 51.1 |
| | PSD [95] | 54.7 |
| | HybridCR [47] | 56.8 |
| | RandLA-Net **+ ERDA** | 63.0 |

Table 4. Results on ScanNet test.

| settings | methods | mIoU |
|----------|---------|------|
| Fully | PointNet [56] | 23.7 |
| | PointNet++ [57] | 32.9 |
| | RandLA-Net [34] | 52.7 |
| | KPConv [68] | 57.6 |
| | LCPFormer [36] | 63.4 |
| | RandLA-Net **+ ERDA** | 64.7 |
| 0.1% | SQN [32] | 54.0 |
| | RandLA-Net **+ ERDA** | 56.4 |

Table 5. Results on SensatUrban test.

| methods | 92 | 183 | 366 | 732 | 1464 |
|---------|-----|-----|-----|-----|------|
| FixMatch [64] | 63.9 | 73.0 | 75.5 | 77.8 | 79.2 |
| **+ ERDA** | 73.5 | 74.9 | 78.0 | 78.5 | 80.1 |

Table 6. Results on Pascal dataset.

Impressively, ERDA yields competitive performance against most supervised methods. For instance, with only 1% of labels, it achieves performance better than its stand-alone baselines with full supervision. Such result indicates that the ERDA is more successful than expected in alleviating the lack of training signals, as also demonstrated qualitatively in Fig. 3.

Therefore, we further extend the proposed method to fully-supervised training, *i.e.,* in setting "Fully" in Tab. 2. More specifically, we generate pseudo-labels for all points and regard the ERDA as an auxiliary loss for fully-supervised learning. Surprisingly, we observe non-trivial improvements (+3.7 for RandLA-Net and +3.4 for CloserLook3D) and achieve the state-of-the-art performance of 72.6 (+2.2) in mIoU with PT. We suggest that the improvements are due to the noise-aware learning from ERDA, which gradually reduces the noise during the model learning and demonstrates to be generally effective. Moreover, considering that the ground-truth labels could suffer from the problem of label noise [38, 90, 65], we also hypothesize that pseudo-labels from ERDA learning could stabilize fully-supervised learning and provide unexpected benefits.

We also conduct the 6-fold cross-validation, as reported in Tab. 3. We find our method achieves a leading performance among both weakly and fully-supervised methods, which further validates the effectiveness of our method.

**Results on ScanNet.** ScanNet [16] is an indoor point cloud dataset that covers 1513 training scenes and 100 test scenes with 20 classes. In addition to the common settings, *e.g.,* 1% labels, it also provides official data efficient settings, such as 20 points, where for each scene there are a pre-defined

| | ER | DA | mIoU | ent. |
|---|---|---|---|---|
| baseline | | | 59.8 | - |
| + pseudo-labels | | | 63.3 | 2.49 |
| | ✓ | | 65.1 | 2.26 |
| + ERDA | | ✓ | 66.1 | 2.44 |
| | ✓ | ✓ | 67.2 | 2.40 |

(a) **ERDA** improves the results and reduces the entropy (ent.), individually and jointly.

| $L_{DA} \setminus \lambda$ | 0 | 1 | 2 |
|---|---|---|---|
| - | - | 65.1 | 66.3 |
| $KL(\mathbf{p}\|\mathbf{q})$ | 66.1 | 67.2 | 66.6 |
| $KL(\mathbf{q}\|\mathbf{p})$ | 66.1 | 65.9 | 65.2 |
| $JS$ | 65.2 | 65.4 | 65.1 |
| $MSE$ | 66.0 | 66.2 | 66.1 |

(b) **ER and DA** provide better results when taking $KL(\mathbf{p}\|\mathbf{q})$ with $\lambda = 1$.

| $k$ | one-hot | soft | ERDA |
|---|---|---|---|
| 64 | 63.1 | 62.8 | 64.5 |
| 500 | 63.0 | 62.3 | 64.1 |
| 1e3 | 63.3 | 63.5 | 65.5 |
| 1e4 | 63.0 | 62.9 | 65.6 |
| dense | 62.7 | 62.6 | 67.2 |

(c) **ERDA** consistently benefits the model with more pseudo-labels ($k$).

Table 7. Ablations on ERDA. If not specified, the model is RandLA-Net trained with ERDA as well as dense pseudo-labels on S3DIS under the 1% setting and reports in mIoU. Default settings are marked in ▢ gray .

set of 20 points with the ground truth label. We evaluate on both settings and report the results in Tab. 4. We largely improve the performance under 0.1% and 1% labels. In 20pts setting, we also employ a convolutional baseline (CloserLook3D) for a fair comparison. With no modification on the model, we surpass MIL-transformer [87] that additionally augments the backbone with transformer modules and multi-scale inference. Besides, we apply ERDA to baseline under fully-supervised setting and achieve competitive performance. These results also validate the ability of ERDA in providing effective supervision signals.

**Results on SensatUrban.** SensatUrban [33] is an urban-scale outdoor point cloud dataset that covers the landscape from three UK cities. In Tab. 5, ERDA surpasses SQN [32] under the same 0.1% setting as well as its fully-supervised baseline, and also largely improves under full supervision. It suggests that our method can be robust to different types of datasets and effectively exploits the limited annotations as well as the unlabeled points.

**Generalizing to 2D Pascal.** As our ERDA does not make specific assumptions on the 3D data, we explore its potential in generalizing to similar 2D settings. Specifically, we study an important task of semi-supervised segmentation on image [72, 89, 8] and implement our method to the popular baseline, FixMatch [64], which combines the pseudo-labels with weak-to-strong consistency and is shown to benefit from stronger augmentation [89]. We use DeepLabv3+ [9] with ResNet-101 [30].

As in Tab. 6, we show that ERDA brings consistent improvement from low to high data regime, despite the existence of strong data augmentation and the very different data as well as setting[4]. It thus indicates the strong generalization of our method. We also see that the improvement is less significant than the 3D cases. It might be because 2D data are more structured (*e.g.,* pixels on a 2D grid) and are thus less noisy than the 3D point cloud.

## 4.3 Ablations and Analysis

We mainly consider the 1% setting and ablates in Tab. 7 to better investigate ERDA and make a more thorough comparison with the current pseudo-label generation paradigm. For more studies on hyper-parameters, please refer to the supplementary.

**Individual effectiveness of ER and DA.** To validate our initial hypothesis, we study the individual effectiveness of $L_{ER}$ and $L_{DA}$ in Tab. 7a. While the pseudo-labels essentially improve the baseline performance, we remove its label selection and one-hot conversion when adding the ER or DA term. We find that using ER alone can already be superior to the common pseudo-labels and largely reduce the entropy of pseudo-labels (ent.) as expected, which verifies that the pseudo-labels are noisy, and reducing these noises could be beneficial. The improvement with the DA term alone is even more significant, indicating that a large discrepancy is indeed existing between the pseudo-labels and model prediction and is hindering the model training. Lastly, by combining the two terms, we obtain the ERDA that reaches the best performance but with the entropy of its pseudo-labels larger than ER only and smaller than DA only. It thus also verifies that the DA term could be biased to uniform distribution and that the ER term is necessary.

**Different choices of ER and DA.** Aside from the analysis in Sec. 3.2, we empirically compare the results under different choices of distance for $L_{DA}$ and $\lambda$ for $L_{ER}$. As in Tab. 7b, the outstanding result justifies the choice of $KL(\mathbf{q}\|\mathbf{p})$ with $\lambda = 1$. Additionally, all different choices and combina-

---

[4]We note that the number in the first row in Tab. 6 denotes the number of labeled images, which is different from the setting of sparse labels in 3D point clouds.

tions of ER and DA terms improve over the common pseudo-labels (63.3), which also validates the general motivation for ERDA.

**Ablating label selection.** We explore in more detail how the model performance is influenced by the amount of exploitation on unlabeled points, as ERDA learning aims to enable full utilization of the unlabeled points. In particular, we consider three pseudo-labels types: common one-hot pseudo-labels, soft pseudo-labels ($\mathbf{p}$), and soft pseudo-labels with ERDA learning. To reduce the number of pseudo-labels, we select sparse but high-confidence pseudo-labels following a common top-$k$ strategy [94, 53] with various values of $k$ to study its influence. As in Tab. 7c, ERDA learning significantly improves the model performance under all cases, enables the model to consistently benefit from more pseudo-labels, and thus neglects the need for label selection such as top-$k$ strategy, as also revealed in Fig. 1. Besides, using soft pseudo-labels alone can not improve but generally hinders the model performance, as one-hot conversion may also reduce the noise in pseudo-labels, which is also not required with ERDA.

## 5  Limitation and Future Work

While we mostly focus on weak supervision in this paper, our method also brings improvements under fully-supervised setting. We would then like to further explore the effectiveness and relationship of $L_{ER}$ and $L_{DA}$ under full supervision as a future work. Besides, despite promising results, our method, like other weak supervision approaches, assumes complete coverage of semantic classes in the available labels, which may not always hold in real-world cases. Point cloud segmentation with missing or novel classes should be explored as an important future direction.

## 6  Conclusion

In this paper, we study the weakly-supervised 3D point cloud semantic segmentation, which imposes the challenge of highly sparse labels. Though pseudo-labels are widely used, label selection is commonly employed to overcome the noise, but it also prevents the full utilization of unlabeled data. By addressing this, we propose a new learning scheme on pseudo-labels, ERDA, that reduces the noise, aligns to the model prediction, and thus enables comprehensive exploitation of unlabeled data for effective training. Experimental results show that ERDA outperforms previous methods in various settings and datasets. Notably, it surpasses its fully-supervised baselines and can be further generalized to full supervision as well as 2D images.

**Acknowledgement.** This project is supported in part by ARC FL-170100117, and IC-190100031.

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

# A   Supplementary: Introduction

In this supplementary material, we provide more details regarding implementation details in Appendix B, more analysis of ERDA in Appendix C, full experimental results in Appendix D, and studies on parameters in Appendix E.

# B   Supplementary: Implementation and Training Details

For the RandLA-Net [34] and CloserLook3D [51] baselines, we follow the instructions in their released code for training and evaluation, which are here (RandLA-Net) and here (CloserLook3D), respectively. Especially, in CloserLook3D[51], there are several local aggregation operations and we use the "Pseudo Grid" (KPConv-like) one, which provides a neat re-implementation of the popular KPConv [68] network (rigid version). For point transformer (PT) [97], we follow their paper and the instructions in the code base that claims to have the official code (here). For FixMatch [64], we use the publicly available implementation here.

Our code and pre-trained models will be released.

# C   Supplementary: Delving into ERDA with More Analysis

Following the discussion in Sec. 3, we study the impact of entropy regularization as well as different distance measurements from the perspective of gradient updates.

In particular, we study the gradient on the score of the $i$-th class *i.e.,* $s_i$, and denote it as $g_i = \frac{\partial L_p}{\partial s_i}$. Given that $\frac{\partial p_j}{\partial s_i} = \mathbb{1}_{(i=j)} p_i - p_i p_j$, we have $g_i = p_i \sum_j p_j (\frac{\partial L_p}{\partial p_i} - \frac{\partial L_p}{\partial p_j})$. As shown in Tab. 8, we demonstrate the gradient update $\Delta = -g_i$ under different situations.

In addition to the analysis in Sec. 3.2, we find that, when $\mathbf{q}$ is certain, *i.e.,* $\mathbf{q}$ approaching a one-hot vector, the update of our choice $KL(\mathbf{p}||\mathbf{q})$ would approach the infinity. We note that this could be hardly encountered since $\mathbf{q}$ is typically also the output of a softmax function. Instead, we would rather regard it as a benefit because it would generate effective supervision on those model predictions with high certainty, and the problem of gradient explosion could also be prevented by common operations such as gradient clipping.

In Fig. 4, we provide visualization for a more intuitive understanding on the impact of different formulations for $L_{DA}$ as well as their combination with $L_{ER}$. Specifically, we consider a simplified case of binary classification and visualize their gradient updates when $\lambda$ takes different values. We also visualize the gradient updates of $L_{ER}$. By comparing the gradient updates, we observe that only $KL(\mathbf{p}||\mathbf{q})$ with $\lambda = 1$ can achieve small updates when $\mathbf{q}$ is close to uniform ($q = \frac{1}{2}$ under the binary case), and that there is a close-0 plateau as indicated by the sparse contour lines.

Additionally, we also find that, when increasing the $\lambda$, all distances, except the $KL(\mathbf{p}||\mathbf{q})$, are modulated to be similar to the updates of having $L_{ER}$ alone; whereas $KL(\mathbf{p}||\mathbf{q})$ can still produce effective updates, which may indicate that $KL(\mathbf{p}||\mathbf{q})$ is more robust to the $\lambda$.

# D   Supplementary: Full Results

We provide full results for the experiments reported in the main paper. For S3DIS [2], we provide the full results of S3DIS with 6-fold cross-validation in Tab. 10. For ScanNet [16] and SensatUrban [33], we evaluate on their online test servers, which are here and here, and provide the full results in Tab. 11 and Tab. 12, respectively.

| $L_{DA}$ | $KL(\mathbf{p}\|\mathbf{q})$ | $KL(\mathbf{q}\|\mathbf{p})$ | $JS(\mathbf{p},\mathbf{q})$ | $MSE(\mathbf{p},\mathbf{q})$ |
|---|---|---|---|---|
| $L_p$ | $H(\mathbf{p},\mathbf{q}) - (1-\lambda)H(\mathbf{p})$ | $H(\mathbf{q},\mathbf{p}) - H(\mathbf{q}) + \lambda H(\mathbf{p})$ | $H(\frac{\mathbf{p}+\mathbf{q}}{2}) - (\frac{1}{2}-\lambda)H(\mathbf{p}) - \frac{1}{2}H(\mathbf{q})$ | $\frac{1}{2}\sum_i (p_i - q_i)^2 + \lambda H(\mathbf{p})$ |
| $g_i$ | $p_i \sum_j p_j(-\log\frac{q_i}{q_j} + (1-\lambda)\log\frac{p_i}{p_j})$ | $p_i - q_i - \lambda p_i \sum_j p_j \log\frac{p_i}{p_j}$ | $p_i \sum_j p_j(\frac{-1}{2}\log\frac{p_i+q_i}{p_j+q_j} + (\frac{1}{2}-\lambda)\log\frac{p_i}{p_j})$ | $p_i(p_i-q_i) - p_i\sum_j p_j(p_j-q_j) - \lambda p_i \sum_j p_j \log\frac{p_i}{p_j}$ |

| Situations | | $\Delta = -g_i$ | | |
|---|---|---|---|---|
| $p_k \to 1$ | $0$ | $q_i - \mathbb{1}_{k=i}$ | $0$ | $0$ |
| $q_1 = ... = q_K$ | $(\lambda-1)p_i\sum_j p_j \log\frac{p_i}{p_j}$ | $\frac{1}{K} - p_i + \lambda p_i \sum_j p_j \log\frac{p_i}{p_j}$ | $p_i\sum_{j\neq i} p_j(\frac{1}{2}\log\frac{Kp_i+1}{Kp_j+1} + (\lambda-\frac{1}{2})\log\frac{p_i}{p_j})$ | $-p_i^2 + p_i \sum_j p_j^2 + \lambda p_i \sum_j p_j \log\frac{p_i}{p_j}$ |
| $q_i \to 1$ | $+\inf$ | $1 - p_i + \lambda p_i \sum_j p_j \log\frac{p_i}{p_j}$ | $p_i\sum_{j\neq i} p_j(\frac{1}{2}\log\frac{p_i+1}{p_j} + (\lambda-\frac{1}{2})\log\frac{p_i}{p_j})$ | $-p_i^2 + p_i(1-p_i) + p_i\sum_j p_j^2 + \lambda p_i \sum_j p_j \log\frac{p_i}{p_j}$ |
| $q_{k\neq i} \to 1$ | $-\inf$ | $-p_i + \lambda p_i \sum_j p_j \log\frac{p_i}{p_j}$ | $p_i\sum_{j\neq i} p_j(\frac{1}{2}\log\frac{p_i}{p_j+\mathbb{1}_{j=k}} + (\lambda-\frac{1}{2})\log\frac{p_i}{p_j})$ | $-p_i^2 - p_i p_k + p_i\sum_j p_j^2 + \lambda p_i \sum_j p_j \log\frac{p_i}{p_j}$ |

Table 8. The formulation of $L_p$ using different functions to formulate $L_{DA}$. We present the gradients $g_i = \frac{\partial L_p}{\partial s_i}$, and the corresponding update $\Delta = -g_i$ under different situations. Analysis can be found in the Sec. 3.2 and Appendix C.

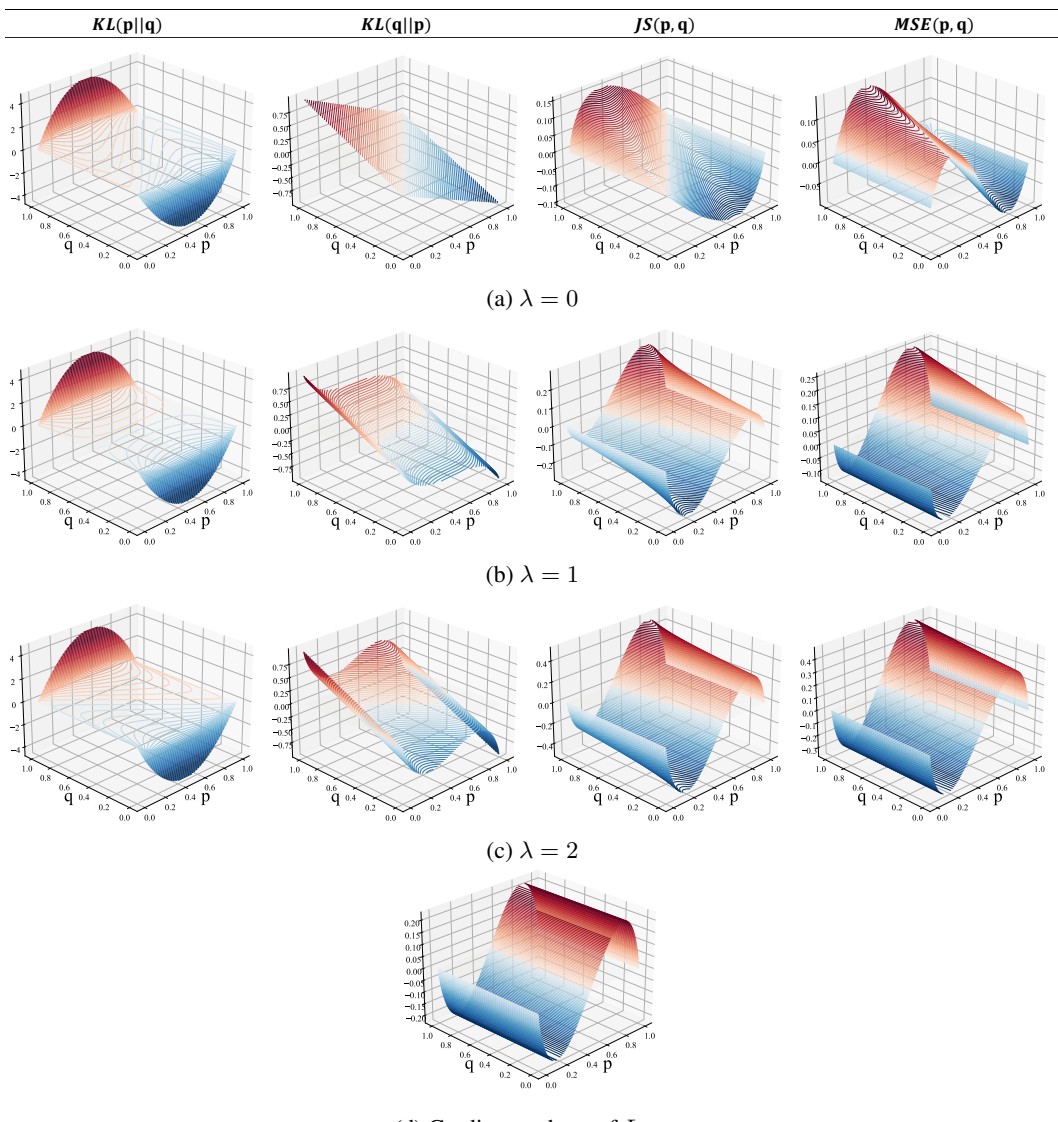

| $KL(\mathbf{p}\|\mathbf{q})$ | $KL(\mathbf{q}\|\mathbf{p})$ | $JS(\mathbf{p},\mathbf{q})$ | $MSE(\mathbf{p},\mathbf{q})$ |
|---|---|---|---|

(a) $\lambda = 0$

(b) $\lambda = 1$

(c) $\lambda = 2$

(d) Gradient updates of $L_{ER}$

Figure 4. Contour visualization of the gradient update with binary classes for better understanding. For a clearer view, we use red for positive updates and blue for negative updates, the darker indicates larger absolute values and the lighter indicates smaller absolute values.

| $m$ | mIoU |
|---|---|
| 0.9 | 66.19 |
| 0.99 | 66.80 |
| 0.999 | 67.18 |
| 0.9999 | 66.22 |

(a) **Momentum update**.

| projection | mIoU |
|---|---|
| - | 65.90 |
| linear | 66.55 |
| 2-layer MLPs | 67.18 |
| 3-layer MLPs | 66.31 |

(b) **Projection network**.

| $\alpha$ | mIoU |
|---|---|
| 0.001 | 65.25 |
| 0.01 | 66.01 |
| 0.1 | 67.18 |
| 1 | 65.95 |

(c) **Loss weight**.

Table 9. Parameter study on ERDA. If not specified, the model is RandLA-Net with ERDA trained with loss weight $\alpha = 0.1$, momentum $m = 0.999$, and 2-layer MLPs as projection networks under 1% setting on S3DIS. Default settings are marked in gray .

# E  Supplementary: Ablations and Parameter Study

We further study the hyper-parameters involved in the implementation of ERDA with the prototypical pseudo-label generation, including loss weight $\alpha$, momentum coefficient $m$, and the use of projection network. As shown in Tab. 9, the proposed method acquires decent performance (mIoUs are all $> 65$ and mostly $> 66$) in a wide range of different hyper-parameter settings, compared with its fully-supervised baseline (64.7 mIoU) and previous state-of-the-art performance (65.3 mIoU by HybridCR [47]).

Additionally, we suggest that the projection network could be effective in facilitating the ERDA learning, which can be learned to specialize in the pseudo-label generation task. This could also be related to the advances in contrastive learning. Many works [10, 11, 25] suggest that a further projection on feature representation can largely boost the performance because such projection decouples the learned features from the pretext task. We share a similar motivation in decoupling the features for ERDA learning on the pseudo-label generation task from the features for the segmentation task.

| settings | methods | mIoU | ceiling | floor | wall | beam | column | window | door | table | chair | sofa | bookcase | board | clutter |
|---|---|---|---|---|---|---|---|---|---|---|---|---|---|---|---|
| | RandLA-Net + ERDA | 71.0 | 94.0 | 96.1 | 83.7 | 59.2 | 48.3 | 62.7 | 73.6 | 65.6 | 78.6 | 71.5 | 66.8 | 65.4 | 57.9 |
| Fully | CloserLook3D + ERDA | 73.7 | 94.1 | 93.6 | 85.8 | 65.5 | 50.2 | 58.7 | 79.2 | 71.8 | 79.6 | 74.8 | 73.0 | 72.0 | 59.5 |
| | PT + ERDA | 76.3 | 94.9 | 97.8 | 86.2 | 65.4 | 55.2 | 64.1 | 80.9 | 84.8 | 79.3 | 74.0 | 74.0 | 69.3 | 66.2 |
| | RandLA-Net + ERDA | 69.4 | 93.8 | 92.5 | 81.7 | 60.9 | 43.0 | 60.6 | 70.8 | 65.1 | 76.4 | 71.1 | 65.3 | 65.3 | 55.0 |
| 1% | CloserLook3D + ERDA | 72.3 | 94.2 | 97.5 | 84.1 | 62.9 | 46.2 | 59.2 | 73.0 | 71.5 | 77.0 | 73.6 | 71.0 | 67.7 | 61.2 |
| | PT + ERDA | 73.5 | 94.9 | 97.7 | 85.3 | 66.7 | 53.2 | 60.9 | 80.8 | 69.2 | 78.4 | 73.3 | 67.7 | 65.9 | 62.1 |

Table 10. The full results of ERDA with different baselines on S3DIS 6-fold cross-validation.

| settings | methods | mIoU | bathtub | bed | books. | cabinet | chair | counter | curtain | desk | door | floor | other | pic | fridge | shower | sink | sofa | table | toilet | wall | wndw |
|---|---|---|---|---|---|---|---|---|---|---|---|---|---|---|---|---|---|---|---|---|---|---|
| Fully | CloserLook3D + ERDA | 70.4 | 75.9 | 76.2 | 77.0 | 68.2 | 84.3 | 48.1 | 81.3 | 62.1 | 61.4 | 94.7 | 52.7 | 19.9 | 57.1 | 88.0 | 75.9 | 79.9 | 64.7 | 89.2 | 84.2 | 66.6 |
| 20pts | CloserLook3D + ERDA | 57.0 | 75.1 | 62.5 | 63.1 | 46.0 | 77.7 | 30.0 | 64.9 | 46.1 | 43.6 | 93.3 | 36.0 | 15.4 | 38.0 | 73.6 | 51.6 | 69.5 | 47.2 | 83.2 | 74.5 | 47.8 |
| 0.1% | RandLA-Net + ERDA | 62.0 | 75.7 | 72.4 | 67.9 | 56.9 | 79.0 | 31.8 | 73.0 | 58.1 | 47.3 | 94.1 | 47.1 | 15.2 | 46.3 | 69.2 | 51.8 | 72.8 | 56.5 | 83.2 | 79.2 | 62.0 |
| 1% | RandLA-Net + ERDA | 63.0 | 63.2 | 73.1 | 66.5 | 60.5 | 80.4 | 40.9 | 72.9 | 58.5 | 42.4 | 94.3 | 50.0 | 35.0 | 53.0 | 57.0 | 60.4 | 75.6 | 61.9 | 78.8 | 73.8 | 62.6 |

Table 11. The full results of ERDA with different baselines on ScanNet [16] test set, obtained from its online benchmark site by the time of submission.

| settings | methods | mIoU | OA | Ground | Vegetation | Buildings | Walls | Bridge | Parking | Rail | Roads | Street Furniture | Cars | Footpath | Bikes | Water |
|---|---|---|---|---|---|---|---|---|---|---|---|---|---|---|---|---|
| Fully | RandLA-Net + ERDA | 64.7 | 93.1 | 86.1 | 98.1 | 95.2 | 64.7 | 66.9 | 59.6 | 49.2 | 62.5 | 46.5 | 85.8 | 45.1 | 0.0 | 81.5 |
| 0.1% | RandLA-Net + ERDA | 56.4 | 91.1 | 82.0 | 97.4 | 93.2 | 56.4 | 57.1 | 53.1 | 5.2 | 60.0 | 33.6 | 81.2 | 39.9 | 0.0 | 74.2 |

Table 12. The full results of ERDA with different baselines on SensatUrban [33] test set, obtained from its online benchmark site by the time of submission.

