# OpenReview forum: "All Points Matter: Entropy-Regularized Distribution Alignment for Weakly-supervised 3D Segmentation"
_NeurIPS.cc/2023/Conference — NeurIPS 2023 poster_

### Official Review · Reviewer_qxVg · 2023-06-22

**Soundness:** 3 good
**Presentation:** 2 fair
**Contribution:** 3 good
**Rating:** 6
**Confidence:** 3

**Summary:**

This submission proposes a semi-supervized pseudo-label-based method for 3D point cloud segmentation method with sparse label annotations. Instead of thresholding on the confidence of pseudo labels, the authors propose to use all unlabelled points and encourage high-confidence pseudo labels by regularizing with their entropy. The authors propose a theoretical analysis of their approach and experiments where they show that their method improves the low-label regime for several backbone and several datasets.

**Strengths:**

- the authors propose a theoretical analysis with some insight

- The experimental results are convincing

- The experiments are extensive: 4 datasets and 4 baselines

**Weaknesses:**


- The entire ER+DA analysis leads to using a loss which is the cross-entropy between the pseudo labels and the prediction, which is already sensible and does not need to be seen as a special case of a more general setting that is never explored anyway. The fact that KL(p,q)+H(q)=CE(p,q) was known and did not need two full pages of motivation and equations (some parts are interesting such as the gradient limits, but would be better suited in the appendix). Equation (6) is completely logical and can be used directly.

- On the other hand, the most interesting part of the paper is the pseudo-label generation hidden in Appendix B, which, as far as the reviewer knows, is novel. The reviewer thinks that the non-backpropagable momentum update of the class prototypes used for pseudo labels makes this method work. Without this, if p could be learned along q, then equation (6) would be meaningless as p=q is a trivial solution, and the pseudo labels would not help at all. The paper would need to be rewritten to highlight this hidden mechanism, brushed in the main paper in a single sentence, and at the method's core.

- Equation (6) reduces to classic cross entropy when all points are labelled, and yet the proposed methods improve the backbones in the fully supervised setting?

- The writing is subpar and lacks rigour. This leads to imprecise or even outright false statements at the core of the motivation. For example, the authors state that regularizing by the entropy decreases the "noise" in pseudo-labels when it actually encourages confident distributions and does not affect noise. There are many vague and uninformative sentences, some dealing with critical aspects of the paper. Some variables are also not introduced. The article remains overall understandable with the suppmat open in another tab.


**Questions:**

Overall, the reviewer believes a very good idea is hidden in the paper; but, it takes a lot of effort to see through the subpar writing, imprecise statements, core ideas hidden in the appendix, and confusing notations.
The reviewer also believes the authors focused on the wrong part of their contribution.
As of now, the paper is not publishable. But if the authors put the work into improving their clarity and rigour, and focused on the important part of their contribution, this could become an impactful paper. The amount of effort might be too much for a rebuttal.

Q1) Equation (6) reduces to classic cross entropy when all points are labelled, and yet the proposed methods improve the backbones in the fully supervised setting?

S1) Swap the derivation of (6) and the pseudo-label generation between the appendix and the main paper.

S2) Add detailed proofs of Table 1 in the appendix

S3) Remove the notion that lower entropy = less noise

**Limitations:**

ok

---

> ### Author Rebuttal · Authors · 2023-08-10
>
> We sincerely thank you for your time and your acknowledgment of our extensive experiments and the provided theoretical analysis. In the following, we address your concerns carefully.
>
> ---
> ### S1. Motivation and swapping contents.
>
> We very much appreciate your confirmation of our idea.
> Besides the formulation of ERDA loss, we also agree that the pseudo-label generation scheme is indeed another important component of our method. However, we believe that the motivation for ERDA loss could be more beneficial for the weakly supervised 3D segmentation task and can provide better insights to the community.
>
> Regarding the motivation, we would like to mention that, some existing papers [3,4,5] also adopt prototypes for pseudo-label generation in this 3D segmentation task.
> Moreover, while methods such as [4,6] also utilize momentum-update for pseudo-label generation, they generally fail to utilize dense pseudo-labels due to the low-quality pseudo-labels.
> This drives us to devise an effective learning scheme of ERDA to tackle the issues of using dense pseudo-labels.
> Therefore, we highlight the design of ERDA in our paper.
>
> Regarding the discussion under general setting, we would also like to mention that, the discussion is related to the mutual learning on pseudo-labeling [1,2], as also discussed in Sec.2 "pseudo-label refinement". While different types of loss are proposed in both 2D and 3D domains [1,2,3,5] during the use of pseudo-labels, there is generally a lack of comparison among these losses.
> We thus motivate the ERDA from a more general setting for better comparison.
>
> Lastly, we will revise our paper and will certainly add more discussions about our pseudo-label generation process, such as by including Fig.3, and will simplify our existing descriptions about motivation to avoid redundancy.
>
> ---
> ### Q1. Improvement for fully supervised learning.
>
> We would like to first clarify that, in fully-supervised setting, we generate the pseudo-labels for all points and regard the ERDA loss as an auxiliary loss for fully-supervised learning.
>
> We would also like to mention that the promising improvement brought by ERDA for the fully supervised setting is truly a surprising benefit as we designed the ERDA primarily for improving learning on unlabelled data.
> Regarding this, we hypothesize that the ERDA for pseudo-labels can help stabilize fully supervised learning.
> Considering that there could be noises in the ground-truth labels [7,8], pseudo-labels from ERDA learning may provide unexpected benefits.
>
> Additionally, we also provided more ablation studies in fully-supervised setting in **Tab.R1** above.
> Interestingly, we find that the distribution alignment (DA) shows more benefits than entropy regularization (ER), but both terms are beneficial.
> We would like to further explore their relationship under full supervision as a promising future work, but it may be beyond the scope of this paper that focuses on weak supervision.
>
> Lastly, we would revise to include more discussion regarding fully-supervised setting in the paper.
>
> ---
> ### S2. Detailed proofs of Table 1
>
> Thanks for your advice and we will revise our paper accordingly.
>
> ---
> ### S3. Notions about entropy and noise.
>
> Regarding the confusion about the connection between "entropy" and "noise" in our paper, we would like to first clarify that we actually consider 'noise' as uncertain predictions in our paper, which we believe aligns with the definition of entropy [9], while this formulation of "noise" does not strictly refer to the general concepts like incorrect predictions. As a result, there may be confusion about connecting "entropy" with "noise" in our paper. We will revise our paper to better clarify these concepts and refine our original statements.
>
> Regarding symbols, some symbols are not defined when introducing the implementation of pseudo-label generation, as they are moved into the appendix due to the limited space. We will revise and include them together with Fig.3 in the main paper for better clarity.
>
>
> ---
> ### References:
> [1] Feng et al. DMT: Dynamic mutual training for semi-supervised learning (Pattern Recognition) \
> [2] Wang et al. Repetitive Reprediction Deep Decipher for Semi-Supervised Learning (AAAI 2020) \
> [3] Zhang et al. Perturbed Self-Distillation: Weakly Supervised Large-Scale Point Cloud Semantic Segmentation (ICCV 2021) \
> [4] Xu et al. Weakly Supervised Semantic Point Cloud Segmentation: Towards 10x Fewer Labels (CVPR 2020) \
> [5] Zhang et al. Weakly supervised semantic segmentation for large-scale point cloud (AAAI 2021) \
> [6] Zhang et al. Semisupervised Momentum Prototype Network for Gearbox Fault Diagnosis Under Limited Labeled Samples (TII 2022) \
> [7] Ye et al. Learning with Noisy Labels for Robust Point Cloud Segmentation (ICCV 2021) \
> [8] Song et al. Learning from Noisy Labels with Deep Neural Networks: A Survey (TNNLS 2022) \
> [9] Shannon. A Mathematical Theory of Communication (1948)

---

> > ### Comment · Reviewer_qxVg · 2023-08-11
> > **Follow-up**
> >
> > Thank you for the clarification.
> >
> > Please make sure to include the positioning of your pseudo-label genetration module with respect ot he work you mention here.
> >
> > The current presentation of the paper seems somewhat oblique. Specifically, the authors only explore the scenario where lambda=1, which leads  the ERDA loss to merely simplify to standard cross-entropy. Given this, I recommend one of two approaches:
> > - Streamline this section to maintain clarity and coherence.
> > - Explore varying lambda values. This would validate the in-depth theoretical discussion and provide readers with a more comprehensive understanding of the benefit of the ERDA loss compared to more standard approaches.
> >
> > Overall, the paper showcases promising ideas, and the results appear commendable. My reservations primarily pertain to its presentation. The emphasis, in my opinion, seems to be on less impactful aspects of an otherwise interesting paper. I've adjusted my rating accordingly.

---

> > > ### Author Response · Authors · 2023-08-12
> > > **Follow-up Response**
> > >
> > > We sincerely thank you for your positive comments and the suggestion that helps refine the quality of this paper.
> > >
> > > We will surely include the pseudo-label generation module with the discussion both here and in the appendix. We hope this could alleviate your concern regarding the balance.
> > >
> > > We will also revise the existing discussion on motivation and general setting to maintain clarity, as also suggested in our previous response to your S1.
> > >
> > > Besides, it is worth mentioning that we have explored the situations when $\lambda$ takes various other values in Tab.7(b). We will refer to these experiments earlier in the discussion to provide a more comprehensive understanding.
> > >
> > > Lastly, we also sincerely thank you for your adjustment on rates.

---

### Official Review · Reviewer_iXj3 · 2023-07-03

**Soundness:** 4 excellent
**Presentation:** 4 excellent
**Contribution:** 3 good
**Rating:** 7
**Confidence:** 4

**Summary:**

This paper proposes two losses for point cloud semantic segmentation. The first one is Entropy Regularization (ER) loss, which makes the pseudo-labels have low entropy and thereby be confident (like one-hot vectors). The other one is Distribution Alignment (DA) loss, which is a KL divergence between the pseudo-labels and the predictions. Extensive experiments show that the combination of these two losses is not only beneficial in a weakly supervised setting but also in a fully supervised setting.

**Strengths:**

1. The proposed method is extremely simple, but theoretically sounding and effective in practice.
2. Extensive experiments strongly support the benefit of the proposed method.
3. Overall presentation is great. Especially, section 3.2 is notably helpful to understand the working logic behind the proposed method.

**Weaknesses:**

1. Could the authors provide a visualization of the (1) entropy map, (2) pseudo-label, and (3) KD loss map of the point cloud, and how they change as training proceeds? The figure would be extremely helpful to elucidate how the proposed method works.

2. Actually, it was quite surprising for me that the proposed method brings performance gain even in a fully-supervised setting. I think more discussion is required for this. Also, it would be great if the authors could provide an ablation study of Table 7(a) in a fully-supervised setting. I expect that DA would be much more beneficial compared to ER in this case, but not sure of course.

3. I understand why the authors refrained from comparing with some super-voxel-based approaches for a fair comparison. However, using a super-voxel partition is one of the widely used settings in 3D semantic segmentation. Hence, to clearly demonstrate the superiority of the proposed method, it would be much better if the authors could provide the performance of ERDA using a super-voxel partition.

4. Some related works are missing.

Joint Learning of 2D-3D Weakly Supervised Semantic Segmentation (NeurIPS 2022)

Box2Mask: Weakly Supervised 3D Semantic Instance Segmentation Using Bounding Boxes (ECCV 2022)

Weakly Supervised 3D Segmentation via Receptive-Driven Pseudo Label Consistency and Structural Consistency (AAAI 2023)



**Questions:**

The results of the experiments on 2D semi-supervised semantic segmentation (SSSS) are interesting. I think the less significant improvement is mainly due to the gap between the weakly-supervised 3D semantic segmentation task and the SSSS task. Maybe the proposed ERDA can make benefit from 2D WSSS (using some points or scribbles).

**Limitations:**

The limitations are properly mentioned in the paper (section 5).

---

> ### Author Rebuttal · Authors · 2023-08-10
>
>
> We sincerely thank you for your positive comments and the overall acknowledgment of our deceptively simple yet effective method. In the following, we address your concerns carefully.
>
> ---
> ### Q1. Visualization of training process.
>
> We thank you for your advice on inspecting the training process through visualization.
> For better understanding, we visualize the ER and DA terms, together with the pseudo-labels in **Fig.R4** above, where we find pseudo-labels can capture meaningful estimation for different semantic classes.
>
> Since KL-Distance could indicate the difference between pseudo-labels and model prediction, we observe that, though pseudo-labels are similar to model prediction in an early stage, pseudo-labels appear to be different from model predictions around complex and cluttered areas as the training proceeds. This may show that the pseudo-labels could capture additional information that helps the model learning.
>
> Moreover, we find pseudo-labels can also provide estimations of different entropy, thus different levels of certainty, for different semantic classes and areas. This may indicate that our pseudo-labels could capture some underlying knowledge of the difficulties of various 3D scenes and classes, which then benefits the model learning.
>
> ---
> ### Q2. Effects of ERDA on fully-supervised learning.
>
> We thank you for your constructive suggestion.
> As shown in **Tab.R1** above, we perform more experiments under the fully-supervised setting for a better investigation.
> Indeed, we find the empirical results follow your expectation that distribution alignment (DA) shows more benefits than entropy regularization (ER), and it indeed further boosts our performance.
> Besides, we may also note that both terms are beneficial compared with the baseline.
>
> Regarding this, we hypothesize that the noise-aware learning of ERDA on pseudo-labels could stabilize fully-supervised learning, considering that ground-truth labels may suffer from the problem of label noise [1,2].
>
> We would also include more discussion regarding full supervision in the paper.
>
> ---
> ### Q3. Comparing with super-voxel methods.
>
> For popular super-voxel methods, such as OTOC [3], during the adaptation of our methods to theirs, we find OTOC requires iterative training, which is both time and resource consuming.
> We could thus not fully reproduce their results.
>
> Instead, since OTOC is based on a popular voxel-based 3D CNN, Minkowvski Unet. We thus adapt our method to its baseline methods (without iterative training). We find that, when using 1% labels, we improve the baseline from 63.4 to 66.8 mIoU by directly adding the proposed ERDA learning without hyper-parameter searching or further adaptation.
>
> Additionally, we would like to first note that, our performance have already surpassed the results of the popular super-voxel methods in weakly supervised point cloud segmentation, such as OTOC[3], under a fair comparison when there is no other dataset-level meta-knowledge available, such as in S3DIS dataset.
> For S3DIS, they provide only "1pt" performance, which is 50.1, while ours is 52.0, using a similar 3D CNN baseline.
>
>
> ---
> ### Q4. Missing related work.
>
> We thank you for your careful read. We also recognize these works are related and would include them in our paper: \
> "Joint 2D-3D" proposes a novel learning setting to leverage paired 2D-3D data to improve weakly-supervised learning in both 2D and 3D domains.\
> "Box2Mask" proposes to use box annotation as weak supervision for 3D instance segmentation, inspired by hough voting and box clustering.\
> "Receptive-Driven" designs three consistency constraints to enhance their pseudo-labels, by utilizing multi-scale consistency, spatial consistency, and semantic consistency.
>
> ---
> ### Q5. Extension to 2D weak-supervised semantic segmentation (WSSS).
>
> We also agree that there could be gaps between the 3D weakly-supervised setting and 2D semi-supervised semantic segmentation (SSSS), especially regarding the form of available labels.
>
> We thank you for your insights and for pointing out a promising future direction about using point-based or scribble-based supervision to make benefits from 2D WSSS.
> Considering the similarity in type of supervision, *e.g.* point-based supervision, we would expect our method to be effective as well. We will explore this direction in the future.
>
> ---
> ### References:
> [1] Ye et al. Learning with Noisy Labels for Robust Point Cloud Segmentation (ICCV 2021)\
> [2] Song et al. Learning from Noisy Labels with Deep Neural Networks: A Survey (TNNLS 2022)\
> [3] Liu et al. One Thing One Click: A Self-Training Approach for Weakly Supervised 3D Semantic Segmentation (CVPR 2021)

---

> > ### Comment · Reviewer_iXj3 · 2023-08-22
> >
> > Thanks for all your effort.
> >
> > All of my concerns are appropriately addressed.
> >
> > I would keep my rating.

---

> > > ### Author Response · Authors · 2023-08-22
> > >
> > > We sincerely thank you for your positive comments as well as your acknowledgment.

---

### Official Review · Reviewer_ZBhv · 2023-07-04

**Soundness:** 3 good
**Presentation:** 3 good
**Contribution:** 3 good
**Rating:** 5
**Confidence:** 4

**Summary:**

This paper proposes a novel learning strategy to regularize the generated pseudo-labels and narrow the gaps between pseudo-labels and model predictions. It introduces an Entropy Regularization loss and a Distribution Alignment loss for weakly supervised learning in 3D segmentation tasks. The approach can better leverage unlabeled data points and achieves state-of-the-art performance under different settings.

**Strengths:**

1. The proposed losses can work with other frameworks to consistently boost their performances. It can potentially achieve better performance with future stronger approaches.
2. The approach can improve many existing approaches by a significant margin.

**Weaknesses:**

I think the motivation and design of the proposed losses needs further analysis.
1. Entropy Regularization loss: does it filter out some high-frequency predictions? How to balance the effects of filtering high-frequency components and removing noise?
2. Distribution Alignment loss:
(a) Does it always improve the performance to encourage the consistency of prediction and pseudo labels?
(b) Are the pseudo labels derived from network predictions? If so, they should be the similar thing. In what cases they tend to be similar and in what cases they are different? This part seems not very clearly presented.

**Questions:**

See above.

**Limitations:**

See above.

---

> ### Author Rebuttal · Authors · 2023-08-10
>
> We sincerely thank you for your acknowledgment of the performance gain of our method and its potential to improve future stronger baselines. In the following, we address your concerns carefully.
>
> ---
> ### Q1. Entropy Regularization (ER) on high-frequency predictions.
>
> We would like to mention that, since the ER works on the soft pseudo-labels at a per-point basis, it would tend to reduce the noise by reducing the occurrence of confusing pseudo-label predictions, *e.g.* predictions with a confidence score around 0.5 in the case of binary classification, rather than spatially smoothing out high-frequency predictions such as areas around scene boundaries and edges.
>
> Although the high-frequency predictions might still be influenced by reducing the level of confusion, we believe that these influences are generally positive. This is because we observed that the edge areas predicted by the model trained with ERDA loss become cleaner and more accurate, as in original Fig.2. We will discuss this more in our paper.
>
> ---
> ### Q2. Questions on Distribution Alignment (DA) term.
>
> Regarding your sub-question **(a)** about whether DA consistently improves, we have performed experiments under different settings to support this, as in Tab.7(a) and Tab.7(b).
>
> In addition, we have also evaluated the effectiveness of DA under fully-supervised setting.
> As shown in **Tab.R1** above, we find DA to be beneficial as well. More interestingly, we notice that the DA term shows even more benefits than the ER term, while both terms are both beneficial.
> We would like to further explore their relationship under full supervision as a promising future work, but it may be beyond the scope of this paper that focuses on weak supervision.
>
> Regarding your sub-question **(b)**, we would like to clarify that the pseudo-labels are generated based on the features and prototypes that are projected by a projection network from the backbone features, as shown in Fig.3 in Appendix B.
> The pseudo-labels and model predictions are thus not necessarily the same or similar.
> For example, we offer the visualization of pseudo-labels, model prediction, and ground-truth labels in Fig.7 of Appendix F. We find that pseudo-labels can provide different estimations and diverge from the model predictions on complex and cluttered areas, such as boards on walls.
>
> Additionally, we also provide a visualization of how pseudo-labels evolve as the training proceeds, as in **Fig.R4** above. Since the KL-Distance (DA term) indicates the similarity between pseudo-labels and model predictions, we find that pseudo-labels gradually learn to be different from the model predictions on cluttered areas, which could regularize the model learning, such as preventing the model from overfitting.
>
> We hope this could shed some light and we will add more discussion and details regarding the pseudo-label generation as well as its comparison with model prediction in our paper.

---

> > ### Comment · Reviewer_ZBhv · 2023-08-20
> > **Thanks for the rebuttal**
> >
> > Thanks for the author's rebuttal. Some of my concerns are addressed and I keep my score as "borderline accept".

---

> > > ### Author Response · Authors · 2023-08-21
> > >
> > > We sincerely thank you for your positive comments.

---

### Official Review · Reviewer_7SFL · 2023-07-06

**Soundness:** 4 excellent
**Presentation:** 3 good
**Contribution:** 3 good
**Rating:** 7
**Confidence:** 4

**Summary:**

This paper considers the task of weakly supervised 3D scene semantic segmentation, where only a limited number of points in each training scene are given labels. Assuming a baseline system that operates within a pseudo-label paradigm, the paper proposes a new set of regularizing loss terms, that aim to (1) reduce pseudo-label entropy and (2) align the distribution of the pseudo-labels and network predictions. Under a default weighting strategy, these terms simplify into cross entropy from the pseudo-labels to the network predictions. Under a wide variety of experimental settings, the paper shows that incorporating this term leads to improved performance for 3d scene semantic segmentation, regardless of the level of supervision.

**Strengths:**

This is a well-written, clear, and compelling paper on an important topic of interest to the community. While the introduced technique is not terribly complex, its benefits are well-justified, and the paper provides substantial analysis to support its inclusion: investigating how gradients from this loss term behave under different prediction settings, and why those gradients align with desirable properties. Further, the paper provides extensive ablation experiments that experimentally support this analysis, and show that all of its components lead to improved performance on the domains under investigation.

The strongest point for the paper is in its thorough and overwhelmingly positive experimental results. For multiple datasets of 3d scenes (all standard), under multiple levels of supervision, adding this loss to an array of baseline models always improves performance, and outperforms previous state-of-the-art models on competitive benchmarks. Substantial improvements are observed when labels are severely limited, and even under fully supervised settings, including this loss term is helpful. From the presented evidence, it seems likely this term should be widely useful for this task and domain in future work, as it presents robustly strong performance under a myriad of framings and settings.


**Weaknesses:**

My biggest outstanding question is to what extent this technique can offer benefits for other domains? In its formulation, there is nothing specific to 3D scene segmentation, so ostensibly it could be generally useful for other weakly supervised domains that employ pseudo-labels. Some evidence is provided that it can transfer to image segmentation, but it would also be interesting to consider domains like 3D shape segmentation. The initial results (on domains other than 3d scenes) provided by the paper are encouraging, but a more thorough analysis would of course strengthen the paper, and likely dramatically improve the reach/impact of the contribution.
Relatedly, I would like to see more analysis / discussion about under what situations this term is helpful? Is it always beneficial to include such a term (no matter the domain / task). For instance, I could imagine that when the initial pseudo-labeling mechanism is highly inaccurate, this term might actually be harmful for learning. For 3D scene segmentation, my prior is that pseudo-labeling techniques are largely successful because strong locality cues in this domain can often be used to propagate labels to nearby unlabeled points, with a relative high degree of confidence; so the quality of initial pseudo-labels for 3D scene segmentation might be higher than would be expected for other domains of interest. It would be interesting to consider the effect that the “goodness” of the pseudo-labeling mechanism has on the final model performance, which could potentially be evaluated in a synthetically designed experiment that introduces “corruption” (at varying levels) into the distributions produced by the pseudo-labeling network. A deeper understanding of how the various components of the system interact with the added loss terms would be beneficial, and may give insight into what other domains and systems may benefit from this insights this paper provides.

Minor:

The formatting of table 2 can be improved. The read highlights are distracting, and largely unneeded as they overstate information. Consider replacing the red text coloring with italics, or better yet, marking only the columns that the baseline does not get improved with the added loss term.


**Questions:**

Perhaps the most surprising result in the paper is that the method improves baselines, even under full label supervision. While I don’t doubt that trend is “real”, as the experimentation seems robust and well-designed, I was not quite satisfied by the explanation given to explain the result on lines 250-254. Is this explanation claiming that the “gt” labels have noise, so using ERD, which is “noise-aware”, can help regulate and remove the noise present in them? If so, this seems like a testable hypothesis (e.g. analyzing differences between pseudo-label predictions and gt label predictions). While I don’t think its required to have a compelte explanation for this phenomenon, the paper should either clarify the explanation here, or simply say that it is unknown why ERDA offers benefits in this paradigm, and to fully understand it would require further study.

**Limitations:**

"Limited" limitations are given, see weaknesses section as to other potential limitations that should be explored.

---

> ### Author Rebuttal · Authors · 2023-08-10
>
> We sincerely thank you for your acknowledgment of both our theoretical and empirical analysis, as well as the potential broader impact. In the following, we address your concerns carefully.
>
> ---
> ### Q1. Application to other tasks.
>
> We also agree that our method could be extended to other tasks, such as the suggested 3D shape segmentation. As in **Tab.R2** above, we find that ERDA also illustrates improvements over the baseline for weakly-supervised shape segmentation.
>
> Along with experiments on semi-supervised image segmentation, these results may indicate that our method can be extended to different domains and tasks.
> In the future, we will investigate a more generic formulation of our method with more complete analysis to benefit other tasks like classification and detection.
>
> ---
> ### Q2: Corruption on pseudo-labels and its relation to model performance.
>
> Thanks for the insightful advice for investigating our methods from the perspective of noise, and we provide related experiments in **Tab.R4** above.
>
> We find that, though corrupted pseudo-labels would affect the model performance, ERDA appears to be relatively robust. Especially, when adding a relatively large noise $\mathcal N(0,0.1)$ on our cosine distance, ERDA is still able to improve the baseline from 59.8 to 65.55 mIoU. Moreover, from the perspective of the final training loss on the segmentation task, the model training is almost not influenced across different noise levels, which may indicate the strong ability of ERDA learning.
>
> ---
> ### Q3: Improvement under full supervision and the potential effect of overcoming potential label noise.
>
> We thank the reviewer for the suggestion to analyze the label noise for a better investigation into the benefits of our method for fully-supervised learning.
>
> While we are also surprised to find our method effective under full supervision,
> we also hypothesize that the ground-truth labels may have noise that would affect the model performance, which has been recognized in general tasks as well as in point cloud datasets [1,2].
>
> More concretely, we checked the difference between the generated pseudo-labels and the ground-truth labels by estimating the accuracy of pseudo-labels *w.r.t.* ground-truth labels.
> The results show that even using the top-5 accuracy of pseudo-labels *w.r.t*. ground-truth labels, the accuracy can only reach 98.6 but not 100. Since the level of uncertainty would be reduced by optimizing with our ERDA loss, this phenomenon partially reveals that the ground-truth labels may have noise that would increase uncertainty and thus affect model training to some extent.
> Regarding this, we acknowledge that such divergence between pseudo-labels and ground-truth data is worth further analysis.
>
> Besides, in **Tab.R1** above, we also perform more ablations under full supervision for a deeper understanding. We find that, with full supervision, while both terms are beneficial, the DA term demonstrates more benefits than ER term, and we observe further improvement on performance.
> We would like to further explore their relationship and how they are effective under full supervision as a promising future work, but it may be beyond the scope of this paper that focuses on weak supervision.
>
> Lastly, we would revise to include more discussion regarding fully-supervised setting in the paper.
>
> ---
> ### Q4: Tab.2 formatting.
> Thanks very much for your advice. We will revise our table accordingly.
>
> ---
> ### References:
> [1] Ye et al. Learning with Noisy Labels for Robust Point Cloud Segmentation (ICCV 2021)\
> [2] Song et al. Learning from Noisy Labels with Deep Neural Networks: A Survey (TNNLS 2022)

---

> > ### Comment · Reviewer_7SFL · 2023-08-14
> >
> > I thank the authors for their detailed and well-written response. I remain very positive on this paper, and would like to see its inclusion to the conference.

---

> > > ### Author Response · Authors · 2023-08-15
> > >
> > > We sincerely thank you for your positive comments and greatly appreciate your acknowledgment.

---

### Official Review · Reviewer_aBL8 · 2023-07-07

**Soundness:** 3 good
**Presentation:** 3 good
**Contribution:** 3 good
**Rating:** 5
**Confidence:** 4

**Summary:**

The paper proposes a novel learning strategy to regularize the generated pseudo-labels and effectively narrow the gaps between pseudo-labels and model predictions, which introduces an Entropy Regularization loss and a Distribution Alignment loss for weakly supervised learning in 3D segmentation tasks, resulting in an ERDA learning strategy.

**Strengths:**

This paper solve an interesting problem, and the experimental results support the conclusions.

**Weaknesses:**

1.There exists some confusions in Figure 1:
a)For (a), there are generally two ways of generating pseudo-labels based on self-training weakly supervised methods. One way is to input both the sample and its augmented version into two different or shared-weight networks (with gradient updating); the other way is to input them into the student network and the teacher network updated via EMA (with only the student network being updated). I think the author's general description of sparse pseudo-labels in (a) is inadequate, which makes it difficult to establish a connection with (b) and understand the essential differences between them.
b)I think the author's naming of (b) is inappropriate. It only optimizes the pseudo-labels p and predictions q simultaneously, without reflecting the "dense" aspect.
2.I appreciate the author's theoretical analysis of entropy regularization and distribution alignment, as well as the evaluation of different loss combinations via formula derivation. However, when I saw Line 160 and Table 7, I wondered if when lambda=1, the best result is achieved, and at this point, the ERDA loss simplifies to a single cross-entropy-based loss that optimizes both p and q. My question is, to my knowledge, I don't think only optimizing p for gradient update at the same time can bring such a high performance gain, and I speculate that the diversity of different perturbations as input enables the model to learn the geometric invariance of features, and the ERDA loss is just the icing on the cake.
3.In Line 207, the author uses a prototypical pseudo-label generation process due to its popularity and simplicity. However, in Table 7(a), the baseline with pseudo-labels already reaches 63.3%. I would like to see the difference between the results of prototypical pseudo-labels and plainest pseudo-labels.
4.I am a little confused about the author's approach under the setting of “fully”. Generally, the PL strategy is applied to unlabeled data, and then leverage prototypes, perturbation or contrastive learning to improve the model's robustness to the unlabeled data. However, in the fully experiment, the author applies ERDA to labeled data, so how can we talk about pseudo-labeling? There is also no gradient update for p.
5.Table 1 is difficult to understand, and I think this part is the core of the method. Therefore, the author needs to provide a detailed description of it.


**Questions:**

See section weakness.

**Limitations:**

There are some issues not explained clearly,  see Section Weakness.

---

> ### Author Rebuttal · Authors · 2023-08-10
>
> We sincerely thank you for your time and efforts and we are grateful for your confirmation of the novelty and effectiveness of the proposed method. In the following, we address your concerns carefully.
>
> ---
> ### Q1. Descriptions in Figure 1:
>
> Thanks for your advice.
>
> For your question **a)**, we would like to mention that the concrete pseudo-label generation pipeline we studied falls in the category of prototypical pseudo-labels, which has been adopted with various modifications in point cloud segmentation [1] and other related fields [2,3]. We will revise our descriptions to make them clearer and more accurate.
>
> Regarding your question **b)**, we agree that the connection between our naming and the 'dense' concept is somewhat implicit.
> In general, we would like to mention that, with our ERDA learning on pseudo-label generation, the benefits of dense pseudo-labels can be better exploited;
> in contrast, existing methods suffer from noises in dense pseudo-labels and require label selections and thus use only limited sparse pseudo-labels.
> We thus highlight the ERDA optimization in Fig.1(b), and only hint on the resulting dense pseudo-labels in Fig.1(b) with corresponding performance comparison in Fig.1(c).
>
> We will revise our figure to improve the clarity of Figure 1 for describing the connection between our method and the 'dense' concept.
>
>
> ---
> ### Q2. Reason for high-performance gain.
>
> We thank you for your acknowledgment of our theoretical analysis.
>
> We would like to first clarify that, as discussed in our paper, optimizing the pseudo-label generation network in a cross-entropy-like loss has the effects of both reducing the entropy of pseudo-labels and maintaining consistency in label distribution at the same time.
> We believe the high performance gain is supported by the effective exploitation of ALL unlabeled data due to the more appropriate utilization of pseudo-labels, which has been demonstrated by our experimental results.
> While other similar formulations (*eg* when $\lambda\neq1$) could also be promising, their improvements are not as significant as our cross-entropy-like ERDA loss, as in Tab.7(b).
>
> Secondly, we would also like to mention that we follow the training of baseline and do not impose any specifically designed perturbation or weak-to-strong consistency.
>
> Lastly, we also demonstrated that the performance would be worsened without using our method when generating dense pseudo-labels, as in Fig.1(c) and Tab.7(c).
> This may also suggest that the diversity of different perturbations would not be helpful and even harmful if not using our ERDA loss.
>
> As a result, we believe our method is far beyond the icing on the cake.
>
> ---
> ### Q3. The difference between prototypical pseudo-labels and plainest pseudo-labels.
>
> By removing the momentum prototype, we further explore the effect of our plainest pseudo-label generation and achieve 62.3 mIoU, which is not significantly different from 63.3 mIoU achieved with momentum prototype in Tab.7(a).
>
> Besides, we may refer to Zhang et al. [1]. Our pseudo-label generation could be closely related to theirs, where they adopt classic prototypical pseudo-labels and can also be viewed as the "plainest" prototypical pseudo-labels with no momentum. They achieve 61.8 mIoU. More concretely, in Fig. 1(c), we have included [1] for comparison with our prototypical pseudo-labels (blue), and it does not yield a significant difference.
>
> ---
> ### Q4. Improvement of ERDA in fully-supervised setting.
>
> We thank you for pointing this out. We would like to first clarify that, in fully-supervised setting, we generate the pseudo-labels for all points and regard the ERDA loss as an auxiliary loss for fully-supervised learning (so there actually exist updates on p).
>
> We would also like to mention that the promising improvement brought by ERDA for the fully supervised setting is truly a surprising benefit as we designed the ERDA primarily for improving learning on unlabelled data. Regarding this, we hypothesize that the ERDA for pseudo-labels can help stabilize fully supervised learning. Considering that there could be noises in the ground-truth labels [4,5], pseudo-labels from ERDA learning may provide unexpected benefits.
>
> To answer this question better, we have also performed more ablations in fully-supervised setting in **Tab.R1** above.
> Interestingly, we find that the distribution alignment (DA) shows more benefits than entropy regularization (ER), but both terms are beneficial.
> We would like to further explore their relationship under full supervision as a promising future work, but it may be beyond the scope of this paper that focuses on weak supervision.
>
> We would also include more descriptions and discussion for fully-supervised setting in the paper.
>
>
> ---
> ### Q5. Better Tab.1 discussion.
>
> We agree that Tab.1 is at the core of our method and thus dedicate Sec 3.2 for discussion. We also realize that prolonged formulas may hinder readability. In addition to the visualization of Tab.1 we offered in Appendix C, we will also revise the paper to facilitate easier and better understanding. For example, we can include some visualization of Tab.1 in the main paper, as in Appendix C.
>
> ---
> ### References:
> [1] Zhang et al. Weakly supervised semantic segmentation for large-scale point cloud (AAAI 2021) \
> [2] Zhang et al. Semisupervised Momentum Prototype Network for Gearbox Fault Diagnosis Under Limited Labeled Samples (TII 2022) \
> [3] Li et al. MoPro: Webly Supervised Learning with Momentum Prototypes (ICLR 2021) \
> [4] Ye et al. Learning with Noisy Labels for Robust Point Cloud Segmentation (ICCV 2021) \
> [5] Song et al. Learning from Noisy Labels with Deep Neural Networks: A Survey (TNNLS 2022)

---

> ### Comment · Reviewer_aBL8 · 2023-08-20
>
> I appriciate the answer made by the authors, and I keep my original positive score.

---

> > ### Author Response · Authors · 2023-08-21
> >
> > We sincerely thank you for your positive comments.

---

### Author Rebuttal · Authors · 2023-08-10

We sincerely thank all reviewers time and effort in providing feedback.

Here, we provide more experiments and visualization for better analysis and understanding of our paper, including the table **R1**, **R2**, **R3**, and Figure **R4** mentioned below.

---

### Decision · Program_Chairs · 2023-09-21

**Decision:**

Accept (poster)

**Comment:**

All reviews are unanimous is recommending acceptance for the paper. aBL8 requests additional clarifications and the ones provided in the author rebuttal for improved descriptions and fully-supervised setting, also as addressed to 7SFL, iXj3 and qxVg, should be included in a revised version. Clarifications on the pseudo-label generation made in response to ZBhv and qxVg should also be included in a revision. Overall, the meta-reviewer agrees with the reviewer consensus that the paper makes interesting contributions to weakly supervised 3D segmentation and may be accepted.